# The community atmospheric-chemistry box model CAABA/MECCA-4.0

**Rolf Sander**[1]**, Andreas Baumgaertner**[2]**, David Cabrera-Perez**[1]**, Franziska Frank**[3]**, Sergey Gromov**[1,*]**, Jens-Uwe Grooß**[4]**, Hartwig Harder**[1]**, Vincent Huijnen**[5]**, Patrick Jöckel**[3]**, Vlassis A. Karydis**[1,6]**, Kyle E. Niemeyer**[7]**, Andrea Pozzer**[1]**, Hella Riede**[1,**]**, Martin G. Schultz**[6,***]**, Domenico Taraborrelli**[6]**, and Sebastian Tauer**[1]

[1]Air Chemistry Department, Max-Planck Institute of Chemistry, P.O. Box 3060, 55020 Mainz, Germany

[2]Deutsches Zentrum für Luft- und Raumfahrt (DLR), Project Management Agency, 53227 Bonn, Germany

[3]Deutsches Zentrum für Luft- und Raumfahrt (DLR), Institut für Physik der Atmosphäre, Oberpfaffenhofen, 82234 Weßling, Germany

[4]IEK-7, Forschungszentrum Jülich, Jülich, Germany

[5]Royal Netherlands Meteorological Institute (KNMI), De Bilt, the Netherlands

[6]IEK-8, Forschungszentrum Jülich, Jülich, Germany

[7]School of Mechanical, Industrial & Manufacturing Engineering, Oregon State University, Corvallis, Oregon, USA

[*] also at: Institute of Global Climate and Ecology (Roshydromet and RAS), Moscow, Russia

[**] now at: Research and Development, Method Development in Remote Sensing, German Weather Service (DWD), Germany

[***] now at: Jülich Supercomputing Center (JSC), Forschungszentrum Jülich, Jülich, Germany

**Correspondence:** R. Sander (rolf.sander@mpic.de)

**Abstract.** We present version 4.0 of the atmospheric chemistry box model CAABA/MECCA which now includes a number of new features: (i) skeletal mechanism reduction, (ii) the MOM chemical mechanism for volatile organic compounds, (iii) an option to include reactions from the Master Chemical Mechanism (MCM) and other chemical mechanisms, (iv) updated isotope tagging, and (v) improved and new photolysis modules (JVAL, RADJIMT, DISSOC). Further, when MECCA is connected to a global model, the new feature of coexisting multiple chemistry mechanisms (Poly-MECCA/CHEMGLUE) can be used. Additional changes have been implemented to make the code more user-friendly and to facilitate the analysis of the model results. Like earlier versions, CAABA/MECCA-4.0 is a community model published under the GNU General Public License.

## 1 Introduction

MECCA (*M*odule *E*fficiently *C*alculating the *C*hemistry of the *A*tmosphere) is an atmospheric chemistry module that contains a comprehensive chemical mechanism with tropospheric and stratospheric chemistry of both the gas and the aqueous phase. In addition to the basic $HO_x$, $NO_x$, and methane chemistry, it also includes non-methane volatile organic compounds (NMVOCs), halogens (Cl, Br, I), sulfur (S), and mercury (Hg) chemistry. For the numerical integration, MECCA uses the KPP software (Sandu and Sander, 2006).

To apply the MECCA chemistry to atmospheric conditions, MECCA must be connected to a base model via the MESSy (*M*odular *E*arth *S*ubmodel *Sy*stem) interface (Jöckel et al., 2010). This base model can be a complex, 3-dimensional model but it can also be a simple box model. CAABA (*C*hemistry *A*s *A B*oxmodel *A*pplication) is such a box model, simulating the atmospheric environment in which the MECCA chemistry takes place.

A full description of CAABA/MECCA has already been published elsewhere (Sander et al., 2005, 2011a). Here, we only present new features that have been implemented after version 3.0. Section 2 describes all changes related to the chemical mechanism of MECCA. In Sect. 3 we show several new options for calculating photolysis rate coefficients in the model. Section 4 presents new features which are only useful when MECCA is coupled to a global (3-dimensional) base model.

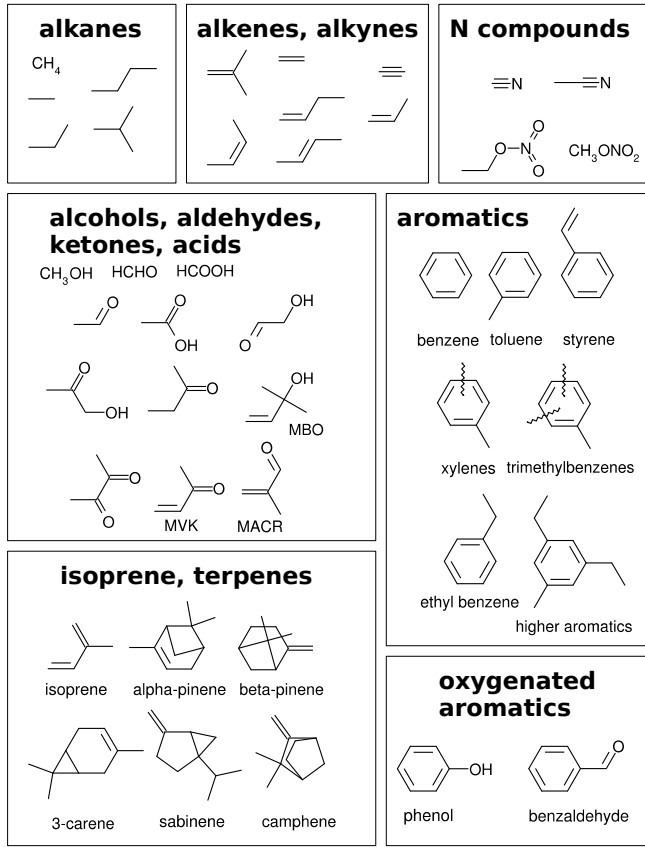

**Figure 1.** Emitted VOCs treated by MOM.

## 2   The chemical mechanism MECCA

MECCA is a chemistry submodel that contains a comprehensive atmospheric reaction mechanism, including 1) the basic $O_3$, $CH_4$, $HO_x$, and $NO_x$ chemistry, 2) NMVOC chemistry, 3) halogen (Cl, Br, I) chemistry, and 4) sulfur chemistry. Recent extensions of MECCA are presented in the following sections.

### 2.1   The Mainz Organic Mechanism (MOM)

The Mainz Organic Mechanism (MOM) is the default oxidation mechanism for volatile organic compounds (VOCs) in MECCA. The current MOM mechanism is a further development of the versions used by Lelieveld et al. (2016) and Cabrera-Perez et al. (2016). It includes developments from Taraborrelli et al. (2012), Hens et al. (2014), and Nölscher et al. (2014). MOM chemistry has been used by Mallik et al. (2018) to study oxidation processes in the Mediterranean atmosphere. Figure 1 shows all 43 emitted species that are treated by MOM. These species are alkanes and alkenes up to four carbon atoms, ethyne (acetylene), two nitriles, isoprene, 2-methyl-3-buten-2-ol (MBO), five monoterpenes, and nine aromatics. Most of the oxidation scheme is explicit. Lump-

ing is used for some isomers with similar properties, e.g., the MOM species "LXYL" presents the sum of $o$-, $m$- and $p$-xylene. All lumped species are marked by the prefix "L" in their names. The full mechanism includes about 600 species and 1600 reactions. A list of all chemical reactions, including rate coefficients and references, is available in the supplement (meccanism.pdf).

The mechanism for the isoprene oxidation was developed starting from MIM2 (Taraborrelli et al., 2009), which is a reduction of MCM v3.1 (Rickard and Pascoe, 2009; Jenkin et al., 1997). The major mechanisms, which regenerate OH under low-$NO_x$ conditions are included. OH-addition to the unsaturated isoprene hydroperoxides has been implemented yielding entirely epoxydiols and OH according to Paulot et al. (2009). The $Z$-1,4- and $Z$-4,1-ISOPO2 isomers undergo 1,6-H-shifts as originally proposed by Peeters et al. (2009). In MOM the corresponding rate coefficients are those computed by Taraborrelli et al. (2012), and the 66% yields of isoprene-derived hydroperoxyenals (HPALDs) are according to Nölscher et al. (2014). For the non-HPALD-yielding channel, the corresponding mechanisms proposed by Peeters et al. (2014) and Jenkin et al. (2015) have been included, however, in a simplified manner. The estimated photo-induced cascade of reactions produces substantial amounts of OH (see Sect. 2.1.5). Finally, methacrolein (MACR) oxidation has been implemented according to Orlando et al. (1999), except for the fate of the methylvinyl radical. The rate of the 1,4-H-shift for the MACRO2 radical is now calculated using the expression reported by Crounse et al. (2012).

Oxidation of the two important terpenes, $\alpha$-pinene and $\beta$-pinene, is based on MCM (Jenkin et al., 2000). However, important modifications following the theoretical work of L. Vereecken have been implemented with some simplifications (Vereecken et al., 2007; Nguyen et al., 2009; Vereecken and Peeters, 2012; Capouet et al., 2008). For instance, minor channels of the OH- and $O_3$-initiated oxidation are neglected.

Aromatics (benzene, toluene, xylenes) are oxidized in the mechanism by Cabrera-Perez et al. (2016), which is to large extent a reduction of the corresponding MCM (Jenkin et al., 2003; Bloss et al., 2005). Photolysis of ortho-nitrophenols yielding HONO has been added according to Bejan et al. (2006) and Chen et al. (2011). Finally, reactions of phenyl peroxy radicals with $NO_2$ yielding $NO_3$ have been added, consistent with Jagiella and Zabel (2007).

Oxidation of VOCs by $O_3$ and $NO_3$ is similar to that in MCM. The oxidation by OH, however, significantly differs from MCM treatment and therefore is detailed in the next section.

### 2.1.1   VOC reactions with OH

Reactions of OH with organic molecules can be either H-abstraction or OH-addition. If available, experimental rate coefficients are preferred and taken mostly from the IUPAC

**Table 1.** SAR parameters and substituent factors in MOM, largely based on Kwok and Atkinson (1995) for H-abstraction and on Peeters et al. (2007) for OH-addition, unless noted otherwise. Most base rate constants and substituent factors are updated with data from Atkinson et al. (2006). Original values for the substituent factors given by Kwok and Atkinson (1995) are listed in parentheses. All rate constants refer to reactions with OH.

| | | $k$ for H-abstraction by OH in $\mathrm{cm^{-3}s^{-1}}$ | |
|---|---|---|---|
| k_p | $k_\mathrm{p}$ (primary) | $4.49 \times 10^{-18} \times (T/\mathrm{K})^2 \times \exp(-320\,\mathrm{K}/T)$ | |
| k_s | $k_\mathrm{s}$ (secondary) | $4.50 \times 10^{-18} \times (T/\mathrm{K})^2 \times \exp(253\,\mathrm{K}/T)$ | |
| k_t | $k_\mathrm{t}$ (tertiary)[a] | $2.12 \times 10^{-18} \times (T/\mathrm{K})^2 \times \exp(696\,\mathrm{K}/T)$ | |
| k_rohro | $k$ (hydroxylic) | $2.1 \times 10^{-18} \times (T/\mathrm{K})^2 \times \exp(-85\,\mathrm{K}/T)$ | |
| k_co2h | $k$ (carboxylic) | $0.7 \times 4.0 \times 10^{-14} \times \exp(850\,\mathrm{K}/T)$ | $= 0.7 \times k_{\mathrm{CH_3CO_2H}}$ |
| k_roohro | $k$ (hydroperoxidic) | $0.6 \times 5.3 \times 10^{-12} \times \exp(190\,\mathrm{K}/T)$ | $= 0.6 \times k_{\mathrm{CH_3OOH}}$ |
| | | Substituent factors $F(X)$ | |
| f_alk | $F(-\mathrm{CH_2}-)$ | 1.23 (1.23) | |
| f_alk | $F(>\mathrm{CH}-)$ | 1.23 (1.23) | |
| f_alk | $F(>\mathrm{C}<)$ | 1.23 (1.23) | |
| f_soh | $F^\mathrm{sec}(-\mathrm{OH})$ | 3.44 (3.50) | $(k_{\mathrm{CH_3CH_2OH \to CH_3CHOH}})/k_\mathrm{s}$ |
| f_toh | $F^\mathrm{tert}(-\mathrm{OH})$ | 2.68 (3.50) | $\dfrac{k_{\text{2-propanol}} - 2k_\mathrm{p} - k_{\mathrm{ROH \to RO}}}{k_{\text{2-methylpropane}} - 3k_\mathrm{p}}$ |
| f_sooh | $F^\mathrm{sec}(-\mathrm{OOH})$ | 8.00 (−) | $(k_{\mathrm{CH_3OOH \to CH_2OOH}})/k_\mathrm{p}$ |
| f_tooh | $F^\mathrm{tert}(-\mathrm{OOH})$ | 8.00 (−) | $(k_{\mathrm{CH_3OOH \to CH_2OOH}})/k_\mathrm{p}$ |
| f_ono2 | $F(-\mathrm{ONO_2})$ | 0.04 (0.04) | |
| f_ch2ono2 | $F(-\mathrm{CH_2ONO_2})$ | 0.20 (0.20) | |
| f_cpan | $F(-\mathrm{C(O)OONO_2})$ | 0.25 (−) | $(k_{\mathrm{CH_3C(O)OONO_2}})/k_\mathrm{p}$ |
| f_allyl | $F^\mathrm{sec}(-\mathrm{allyl})$ | $3.6^b$ (1.00) | $\dfrac{k_{\mathrm{CH_2CHCH_3 \to CH_2CHCH_2}}}{k_{\mathrm{CH_3CH_2CH_3 \to CH_3CH_2CH_2}}}$ |
| f_cho | $F(-\mathrm{CHO})$ | 0.55 (0.75) | $\dfrac{k_{\mathrm{HOCH_2CHO \to HOCHCHO}}}{k_\mathrm{p} F^\mathrm{sec}(-\mathrm{OH})}$ |
| f_co2h | $F(-\mathrm{COOH})$ | 1.67 (0.74) | $(k_{\mathrm{CH_3COOH \to CH_2COOH}})/k_\mathrm{p}$ |
| f_co | $F(-\mathrm{C(=O)R})$ | 0.73 (0.75) | $(k_{\mathrm{CH_3CHO \to CH_3CO}})/k_\mathrm{t}$ |
| f_o | $F(=\mathrm{O})$ | 8.15 (8.70) | $(k_{\mathrm{CH_3CHO \to CH_3CO}})/k_\mathrm{t}$ |
| f_pch2oh | $F^\mathrm{prim}(-\mathrm{CH_2OH})$ | 1.29 (1.23) | $(k_{\mathrm{CH_3CH_2OH \to CH_2CH_2OH}})/k_\mathrm{p}$ |
| f_tch2oh | $F^\mathrm{tert}(-\mathrm{CH_2OH})$ | 0.53 (−) | $(k_{\mathrm{HOCH_2CHO \to HOCH_2CO}})/(k_\mathrm{t}F(=\mathrm{O}))$ |
| | | $k$ for OH-addition to double bonds in $\mathrm{cm^{-3}s^{-1}}$ | |
| k_adp | $k_\mathrm{adp}$ (primary) | $4.5 \times 10^{-12} \times (T/300\,\mathrm{K})^{-0.85}$ | $0.5 k_{\mathrm{C_2H_4}}$ (high pressure limit) |
| k_ads | $k_\mathrm{ads}$ (secondary) | $1/4 \times (1.1 \times 10^{-11} \times \exp(485\,\mathrm{K}/T)$ $+ 1.0 \times 10^{-11} \times \exp(553\,\mathrm{K}/T))$ | $0.5 k_{\text{cis/trans-2-butene}}$ |
| k_adt | $k_\mathrm{adt}$ (tertiary) | $1.922 \times 10^{-11} \times \exp(450\,\mathrm{K}/T) - k_\mathrm{ads}$ | $k_{\text{2-methyl-2-butene}} - k_\mathrm{ads}$ |
| k_adsecprim | | $3.0 \times 10^{-11}$ | $0.5(k_{\text{1,3-butadiene}} - 2k_\mathrm{adp})$ |
| k_adtertprim | | $5.7 \times 10^{-11}$ | $0.5(k_{\text{2,3-dimethyl-1,3-butadiene}} - 2k_\mathrm{adp})$ |
| | | Substituent factors $F_a(X)$ | |
| a_pan | $F_a(-\mathrm{C(O)OONO_2})$ | 0.56 (−) | $k_{\mathrm{MPAN}}/k_{\text{2-methylpropene}}$ |
| a_cho | $F_a(-\mathrm{CHO})$ | 0.31 (0.34) | $k_{\text{methacrolein}}^\mathrm{add}/k_{\text{2-methylpropene}}$ |
| a_coch3 | $F_a(-\mathrm{C(O)CH_3})$ | 0.76 (0.90) | $k_{\mathrm{MVK}}/k_{\text{propene}}$ |
| a_ch2oh | $F_a(-\mathrm{CH_2OH})$ | 1.7 (1.6) | $k_{\text{2-propene-1-ol}}/k_{\text{propene}}$ |
| a_ch2ooh | $F_a(-\mathrm{CH_2OOH})$ | 1.7 (−) | $k_{\text{2-propene-1-ol}}/k_{\text{propene}}$ |
| a_coh | $F_a(>\mathrm{CHOH})$ | 2.2 (1.6) | $k_{\text{1-pentene-3-ol}}/k_{\text{1-pentene}}$ |
| a_cooh | $F_a(>\mathrm{CHOOH})$ | 2.2 (1.6) | $k_{\text{1-pentene-3-ol}}/k_{\text{1-pentene}}$ |
| a_co2h | $F_a(-\mathrm{C(O)OH})$ | 0.25 (0.25) | |
| a_ch2ono2 | $F_a(-\mathrm{CH_2ONO_2})$ | 0.64 (0.47) | $\dfrac{k_{\mathrm{O_2NOCH_2C(CH_3)=CHCH_2OH}}}{F_a(-\mathrm{CH_2OH})k_{\text{2-methyl-2-butene}}}$ |

[a] There is a sign error in Kwok and Atkinson (1995) who present the value $\exp(-696\,\mathrm{K}/T)$ instead of $\exp(696\,\mathrm{K}/T)$.
[b] Median value from the range calculated by Vereecken and Peeters (2001).

kinetic data evaluation (Atkinson et al., 2006; Wallington et al., 2018). Unmeasured rate coefficients for the $C_1$ to $C_5$ species are estimated with a site-specific Structure-Activity Relationship (SAR) similar to MCM, based on the work of Atkinson (1987) and Kwok and Atkinson (1995). The base rate coefficients for OH-addition to double bonds are taken from the more recent SAR by Peeters et al. (2007). For the $C_6$ to $C_{11}$ closed-shell species, the MCM rate coefficients are retained. It is worth noting that the SAR-estimated ones have no temperature-dependence and are only given at 298 K. The effect of neighbouring groups is expressed by substituent factors and is differentiated by functional group. Most substituent factors by Kwok and Atkinson (1995) are updated or calculated *ex novo* by computing the relative rate coefficient of OH with the simplest VOC bearing the substituent relative to the one of its parent compound (Tab. 1). A clear limitation of this approach is that for OH-addition no substituent effect on the branching ratios is considered. No rigorous evaluation of the SAR has been conducted and the estimation uncertainty is expected to be in the same range as for the SAR used by MCM.

The general formulae for H-abstraction by OH are:

$$k(\mathrm{CH_3}X) = k_\mathrm{p} \cdot F(X) \tag{1}$$
$$k(\mathrm{CH_2}XY) = k_\mathrm{s} \cdot F(X) \cdot F(Y) \tag{2}$$
$$k(\mathrm{CH}XYZ) = k_\mathrm{t} \cdot F(X) \cdot F(Y) \cdot F(Z) \tag{3}$$

where $k_\mathrm{p}$, $k_\mathrm{s}$, $k_\mathrm{t}$ are the group rate coefficients for the hydrogens on the primary, secondary and tertiary carbon atoms, respectively, and $F(X)$ is the factor for the substituent $X$.

The SAR for OH-addition to (poly)alkenes is based on the hypothesis that the site-specific rate coefficient depends solely on the stability of the radical product (Peeters et al., 2007). Thus, rate coefficients for the formation of primary, secondary and tertiary radicals are derived from the high-pressure limits for ethene, 2-butene and 2,3-dimethyl-2-butene, respectively. It is worth noting that for the tertiary radical formation, Peeters et al. (2007) used solely the rate coefficient for 2,3-dimethyl-2-butene and not that for 2-methyl-2-butene minus that for the secondary radical.

### 2.1.2 RO₂ reactions with NOₓ and NO₃

Reactions with NO are the dominant sink for $RO_2$ under polluted conditions. The $RO_2$-size independent MCM rate coefficient is used with the exception of $CH_3O_2$ and $CH_3CH_2O_2$, for which the IUPAC recommendations are followed (Atkinson et al., 2006). In general, the two possible reaction channels are considered:

$$\mathrm{RO_2 + NO} \rightarrow (1-\alpha) \times (\mathrm{RO + NO_2}) \tag{R1}$$
$$\alpha \times \mathrm{RONO_2} \tag{R2}$$

with $\alpha$ being the alkyl nitrate yield for the formation of alkyl nitrates, which curb tropospheric ozone production. Acyl $RO_2$ do not form nitrates. The $CH_3ONO_2$-yield is calculated according to Butkovskaya et al. (2012) with a reduction

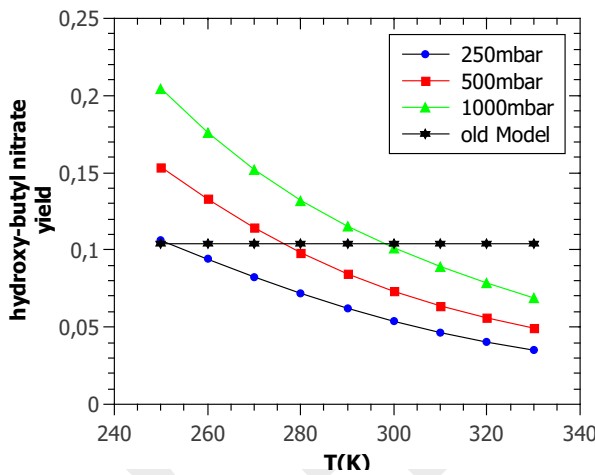

**Figure 2.** Temperature- and pressure-dependent nitrate yield for the secondary hydroxybutyl peroxy radical obtained calculated by MOM. A constant yield of about 10 % ("old model") is used by MCM.

according to Flocke et al. (1998). The $CH_3CH_2ONO_2$-yield is calculated according to Butkovskaya et al. (2010). For all other peroxy radicals the corresponding alkyl nitrate yields are calculated with the relationship by Arey et al. (2001), which depends on temperature, pressure and molecular size. However, the latter is represented not by the number of carbon atoms but by the number of heavy atoms (excluding the $-OO$ moiety) according to Teng et al. (2015). The oxygen atom in $\beta$-carbonyl $RO_2$ is not counted. Due to disagreement in the literature, no dependence of $\alpha$ on the degree of $RO_2$ substitution (primary, secondary and tertiary) is considered. Reduction factors for $\beta$- and $\gamma$-carbonyl $RO_2$ are derived from Praske et al. (2015) and for bicyclic $RO_2$ from aromatics are derived from Elrod (2011). As an example, Figure 2 shows the predicted variable yield for the nitrate of the secondary hydroxy butyl peroxy radical.

Formation and decomposition of many peroxy nitrates is considered. The equilibria of acyl peroxy nitrates with their parent $RO_2$ are represented as in MCM but the JPL kinetic data (Burkholder et al., 2015) is used. Only three alkyl peroxy nitrates, $CH_3O_2NO_2$, $CH_3CH_2O_2NO_2$ and $CH_3COCH_2O_2NO_2$, are represented. The equilibrium reactions for the latter are taken from Tyndall et al. (2001), Sehested et al. (1998) and Kirchner et al. (1999). Reactions of peroxy radicals with $NO_3$ all produce the corresponding alkoxy radical and $NO_2$:

$$\mathrm{RO_2 + NO_3} \rightarrow \mathrm{RO + NO_2 + O_2} \tag{R3}$$

The temperature-independent rate coefficient of $k(\mathrm{C_2H_5O_2 + NO_3}) = 2.3 \times 10^{-12} \ \mathrm{cm^{-3}s^{-1}}$ is used for all $RCH_2O_2$. For acyl peroxy radicals, an enhancement factor of $k(\mathrm{CH_3C(O)OO + NO_3})/k(\mathrm{C_2H_5O_2 + NO_3}) = 1.74$ is calculated based on the peroxy acetyl radical.

### 2.1.3 RO$_2$ reactions with HO$_x$

HO$_2$ reactions are often competitive with NO reactions of peroxy radicals. The former reactions are known to proceed via three channels

$$RO_2 + HO_2 \quad \rightarrow \quad RO + OH + O_2 \qquad (R4)$$
$$ROOH + O_2 \qquad (R5)$$
$$ROH + O_3 \qquad (R6)$$

of which only the first is a radical propagating channel. Alkyl peroxy radicals cannot have the O$_3$-channel and their rate coefficient is calculated as a function of the number of carbons according to the fitting formula provided by Saunders et al. (2003) and Boyd et al. (2003). The branching ratios of the OH-channel for $\beta$-carbonyl, alkoxy and bicyclic peroxy radicals are taken from Dillon and Crowley (2008), Orlando and Tyndall (2012) and Birdsall et al. (2010), respectively. A 10 % OH-yield for reactions of $\beta$-hydroxyl peroxy radicals is taken from the isoprene oxidation study of Liu et al. (2013), which is consistent with the results of Groß (2013) and Paulot et al. (2009). The HO$_2$ reaction of the simplest acyl peroxy radical (CH$_3$CO$_3$) has unique branching ratios as determined by direct OH and O$_3$ measurements (Groß et al., 2014). For all other acyl peroxy radicals the kinetic data for $\beta$-hydroxy acyl peroxy radicals, e.g. HOCH$_2$CO$_3$, are taken from Groß (2013) with the rate coefficient having the temperature dependence as recommended by IUPAC.

There is laboratory evidence for a non-negligible reaction of CH$_3$O$_2$ with OH (Bossolasco et al., 2014):

$$CH_3O_2 + OH \quad \rightarrow \quad CH_3O + HO_2 \qquad (R7)$$

The lower limit of the rate coefficient $1.4 \times 10^{-10}\,\mathrm{cm^{-3}s^{-1}}$ reported by Bossolasco et al. (2014) is used in MOM. This is consistent with the revised experimental value by the same lab (Assaf et al., 2016). The major reaction channel involving HO$_2$ elimination represents $(80 \pm 20)$ % and is set as the only channel (Assaf et al., 2017). The other possible channels are very uncertain and are therefore not included.

### 2.1.4 RO$_2$ permutation reactions

The self and cross reactions of organic peroxy radicals are treated according to the permutation reaction formalism in MCM (Jenkin et al., 1997). Every organic peroxy radical reacts in a pseudo-first-order reaction with a rate coefficient that is expressed as

$$k^{1st} = 2 \times \sqrt{k_{RO_2} \times k_{CH_3O_2}} \times [RO_2] \qquad (4)$$

where $k_{RO_2}$ = second-order rate coefficient of the self reaction of the organic peroxy radical, $k_{CH_3O_2}$ = second-order rate coefficient of the self reaction of CH$_3$O$_2$, and [RO$_2$] = sum of the concentrations of all organic peroxy radicals. The formalism is a simplification of the approach by Madronich

and Calvert (1990) under the assumption that the dominant co-reactant of RO$_2$ is CH$_3$O$_2$. The value of $k_{CH_3O_2}$ is taken from the IUPAC recommendations. Expressions for $k_{RO_2}$ distinguish acyl from alkyl peroxy radicals. The latter are differentiated by the degree and kind of substituents close the $-$OO moiety. The rate expressions (Tab. 2) are not from MCM, except for $\beta$-hydroxyl radicals, and have a temperature dependence.

### 2.1.5 Photo-induced reactions

The enhanced photolysis of carbonyl nitrates from isoprene is implemented according to Barnes et al. (1993) and Müller et al. (2014). The enhancement is applied to the photolysis rate coefficients ($j$-values) of nitrooxyacetone (NOA), nitrooxyacetaldehyde (NO3CH2CHO), lumped nitrates of methyl ethyl ketone (LMEKNO3), nitrates of MVK and MACR and unsaturated C$_5$-nitrooxyaldehyde from the isoprene + NO$_3$ reaction.

Keto-enol tautomerization of aldehydes induced by light absorption is implemented based on data for acetaldehyde (Clubb et al., 2012). The enols are in equilibrium with the corresponding aldehydes by HCOOH-catalysis (da Silva, 2010). Formic acid is then produced upon reaction of the enols with OH similarly to the simplest enol (So et al., 2014). Vinyl alcohol is also produced in the photolysis of propanal.

Photolysis of HPALDs is according to Peeters et al. (2014) and Jenkin et al. (2015) and the subsequent photolysis of the resulting carbonyl enols (HVMK and HMAC) is treated according to Nakanishi et al. (1977) and Messaadia et al. (2015).

Nitrophenols undergo photolysis yielding HONO, according to Bejan et al. (2006) and Chen et al. (2011), and assumed co-products being cyclic ketenes. However, the OH-formation channel (Cheng et al., 2009; Vereecken et al., 2016) is not implemented.

Conjugated unsaturated dialdehydes like butenedial and 2-methyl-butenedial from isoprene and aromatics oxidations undergo photolysis based on Xiang et al. (2007) and MCM. Only the major channel, CO loss, is considered, and the $j$-values are scaled with $j(NO_2)$. The ketenes from photolysis of hydroperoxyacetyl conjugated unsaturated aldehydes from isoprene, conjugated unsaturated dialdehydes and nitrophenols undergo photo-dissociation yielding CO and an excited Criegee intermediate. The $j$-value is assumed to be the same as that for MVK with a unity quantum yield.

## 2.2 Other chemical mechanisms

In addition to the native chemistry mechanism of MECCA (available in the file gas.eqn), several other, independent mechanisms are now provided as well. The chemical mechanisms CB05BASCOE and MOZART from the Copernicus Atmosphere Monitoring Service project (CAMS 42), and the Jülich Atmospheric Mechanism (JAM002) have been con-

**Table 2.** Second-order rate constants $k^{2nd}$ for permutation reactions (in $cm^{-3}s^{-1}$). Here, $k_{CH_3O_2} = 1.03E\text{-}13 \times \exp(365\,K/T)\,cm^{-3}s^{-1}$ is for the self-reaction of $CH_3O_2$.

| variable | $k^{2nd} = 2 \times \sqrt{k_{RO_2} \times k_{CH_3O_2}}$ | based on | reference |
|---|---|---|---|
| k_RO2RCO3 | $2 \times 2\text{E-}12 \times \exp(500\,K/T)$ | $CH_3CO_3 + CH_3O_2$ | Atkinson et al. (2006) |
| Alkyl radicals (unsubstituted, $> C_3$) | | | |
| k_RO2pRO2 | $2 \times \sqrt{1\text{E-}12 \times k_{CH_3O_2}}$ | $RO_2 = (CH_3)_2CHCH_2O_2$ | Glover and Miller (2005) |
| k_RO2sRO2 | $2 \times \sqrt{1.6\text{E-}12 \times \exp(-2200\,K/T) \times k_{CH_3O_2}}$ | $RO_2 = \text{i-}C_3H_7O_2$ | Orlando and Tyndall (2012) |
| k_RO2tRO2 | $2 \times 3.8\text{E-}13 \times \exp(-1430\,K/T)$ | $\text{t-}C_4H_9O_2 + CH_3O_2$ | Wallington et al. (2018) |
| Alkyl radical with either O or Cl in $\beta$ | | | |
| k_RO2pORO2 | $2 \times 7.5\text{E-}13 \times \exp(500\,K/T)$ | $CH_3COCH_2O_2 + CH_3O_2$ | Orlando and Tyndall (2012) |
| k_RO2sORO2 | $2 \times \sqrt{7.7\text{E-}15 \times \exp(1330\,K/T) \times k_{CH_3O_2}}$ | $RO_2 = CH_3CH(OH)CH(O_2)CH_3$ | Orlando and Tyndall (2012) |
| k_RO2tORO2 | $2 \times \sqrt{4.7\text{E-}13 \times \exp(-1420\,K/T) \times k_{CH_3O_2}}$ | $RO_2 = (CH_3)_2C(OH)CO_2(CH_3)_2$ | Orlando and Tyndall (2012) |
| Allyl- and $\beta$-hydroxy alkyl radicals | | | |
| k_RO2LISOPACO2 | $2 \times \sqrt{(2.8\text{E-}12 + 3.9\text{E-}12)/2 \times k_{CH_3O_2}}$ | $RO_2 = $ ISOPAO2 and ISOPCO2 | Saunders et al. (2003) |
| k_RO2ISOPBO2 | $2 \times \sqrt{6.9\text{E-}14 \times k_{CH_3O_2}}$ | $RO_2 = $ ISOPBO2 | Saunders et al. (2003) |
| k_RO2ISOPDO2 | $2 \times \sqrt{4.8\text{E-}12 \times k_{CH_3O_2}}$ | $RO_2 = $ ISOPDO2 | Saunders et al. (2003) |

verted to KPP format and introduced into MECCA. It is also possible to use our previous, simple mechanism MIM1 (Jöckel et al., 2016). In addition, chemical mechanisms extracted and downloaded from the MCM web page can be converted with a script which makes them compatible with CAABA/MECCA. All mechanisms are suitable for stratospheric as well as tropospheric calculations. They all include the chemistry of chlorine bromine, and isoprene. They differ in the treatment of terpenes. MIM1 has no terpenes at all. CB05BASCOE and MOZART include terpenes as a lumped species. Only MCM, MOM and JAM002 treat some terpenes individually, e.g., pinene. The JAM002 mechanism is larger than CB05BASCOE and MOZART but small compared to MECCA with MOM. The very detailed MCM is the largest of all. More information about the chemical mechanisms is provided in the following sections.

### 2.2.1 CB05BASCOE

The CB05BASCOE scheme (Huijnen et al., 2016) is a merge of a tropospheric and stratospheric chemistry scheme. The tropospheric chemistry is based on the Carbon Bond mechanism 2005 (CB05, Yarwood et al., 2005). Here, a lumping approach is adopted for organic species by defining a separate tracer species for specific types of functional groups. The scheme has been modified and extended to include an explicit treatment of $C_1$ to $C_3$ species (Williams et al., 2013), $SO_2$, dimethyl sulfide (DMS), methyl sulfonic acid (MSA) and ammonia ($NH_3$), as described by Huijnen et al. (2010). The reaction rates follow the recommendations given in either the JPL or IUPAC evaluation (Burkholder et al., 2015; Wallington et al., 2018). The stratospheric chemistry is based on that from the BASCOE (Belgian Assimilation System for Chemical ObsErvations) system (Errera et al., 2008) and is labelled "sb15b". This chemical scheme merges the reac-

tion lists developed by Errera and Fonteyn (2001) to produce short-term analyses, with the list included in the SOCRATES 2-D model for long-term studies of the middle atmosphere (Brasseur et al., 2000; Chabrillat and Fonteyn, 2003). The list of species includes all the ozone-depleting substances and greenhouse gases necessary for multi-decadal simulations of the couplings between dynamics and chemistry in the stratosphere, as well as the reservoir and short-lived species necessary for a complete description of stratospheric ozone photochemistry. Gas-phase and heterogeneous reaction rates are taken from the JPL evaluations 17 and 18 (Sander et al., 2011b; Burkholder et al., 2015). The merged reaction mechanism includes 99 species interacting through 211 gas-phase and 10 heterogeneous reactions. Details regarding its implementation and evaluation within the ECMWF Integrated Forecasting System (IFS) are given by Huijnen et al. (2016).

### 2.2.2 MOZART

The tropospheric chemistry in MOZART (Model of OZone And Related Tracers) is based on the MOZART-3 mechanism by Kinnison et al. (2007). It includes additional species and reactions from MOZART-4 (Emmons et al., 2010) and further updates from the Community Atmosphere Model with interactive chemistry, referred to as CAM4-chem (Lamarque et al., 2012). The chemical mechanism includes an updated isoprene oxidation scheme and a better treatment of volatile organic compounds, with lumped species to represent large alkanes, alkenes and aromatic compounds as well as their oxidation products. Overall, it includes the degradation of $C_1$, $C_2$, $C_3$, $C_4$, $C_5$, $C_7$, and $C_{10}$ species. The heterogeneous chemistry in the troposphere is implemented according to the corresponding module from CB05BASCOE. MOZART includes the extended stratospheric chemistry discussed by Kinnison et al. (2007) with further updates from

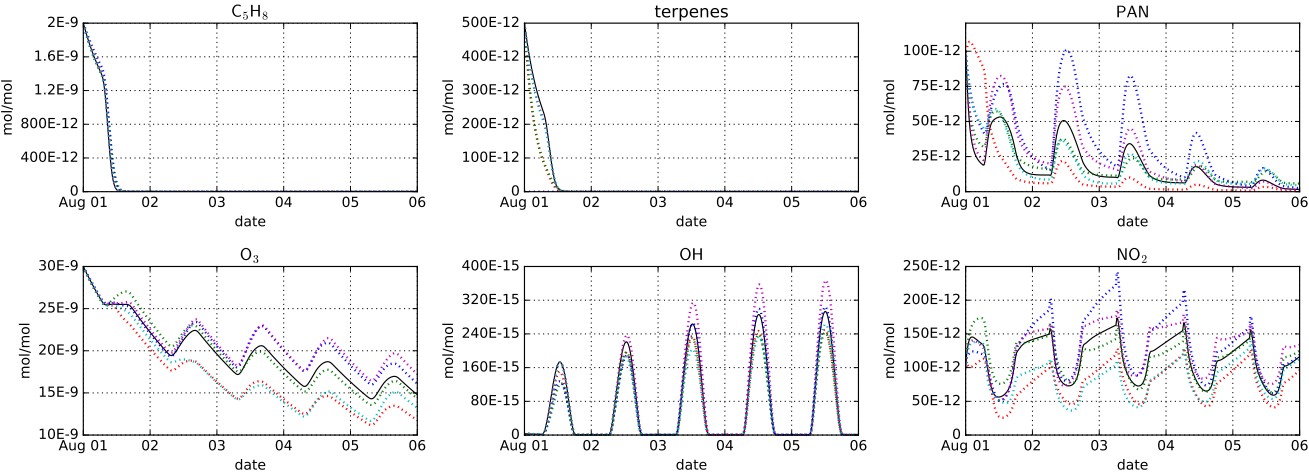

**Figure 3.** Intercomparison of the MOM (black), CB05BASCOE (red), MOZART (green), MIM1 (blue), MCM (magenta), and JAM002 (cyan) mechanisms. The simulations represent the boundary layer over the Amazon forest. They start on 1 August at midnight and last for 5 days. Temperature, pressure, and relative humidity are set to 301 K, 101325 Pa, and 70 %, respectively. The model is initialized with 2 nmol/mol isoprene ($C_5H_8$), 500 pmol/mol of terpenes (MOM: 100 pmol/mol of $\alpha$-pinene, $\beta$-pinene, camphene, carene, and sabinene each; CB05BASCOE and MOZART: lumped terpenes; MIM1: no terpenes; MCM and JAM002: 200 pmol/mol $\alpha$-pinene and 300 pmol/mol $\beta$-pinene), and 100 pmol/mol PAN. During the model simulation, emissions of NO are set to $3.3 \times 10^{-9}$ cm$^{-2}$s$^{-1}$ (Taraborrelli et al., 2009).

CAM4-chem (Lamarque et al., 2012; Tilmes et al., 2016). This includes detailed gas-phase halogen chemistry of chlorine and bromine. The stratospheric chemistry accounts for heterogeneous processes on liquid sulfate aerosols and polar stratospheric clouds, following the approach of Considine et al. (2000). Overall, the MOZART mechanism includes 117 gas-phase species, 65 photolysis and 247 gas-phase reactions. Rate coefficients are taken from the JPL recommendations (Sander et al., 2006, 2011b).

### 2.2.3 JAM002

Version 2 of the Jülich Atmospheric Mechanism (JAM002) has been implemented in the ECHAM-HAMMOZ chemistry-climate model (Schultz et al., 2018). It is a blend of the stratospheric chemistry scheme of the Whole Atmosphere Chemistry Climate Model (WACCM, Kinnison et al., 2007) and version 4 of the tropospheric MOZART model (Emmons et al., 2010). The combined chemistry scheme of WACCM and MOZART has been enhanced with a detailed representation of the oxidation of isoprene following the Mainz Isoprene Mechanism 2 (MIM2, Taraborrelli et al., 2009), and by adding a few primary volatile organic compounds and their oxidation chains. The isoprene oxidation scheme includes recent discoveries of 1,6 H-shift reactions (Peeters et al., 2009), the formation of epoxide (Paulot et al., 2009) and the photolysis of HPALDs (Wolfe et al., 2012). Some of the reaction products and rates were taken from MCM (Jenkin et al., 2015). Radical-radical reactions have been substantially revised since Emmons

et al. (2010). In contrast to MCM, JAM002 does not use a radical pool but instead follows the pathways of peroxy radical reactions with $HO_2$, $CH_3O_2$, and $CH_3COO_2$ (peroxy acetyl) as explicitly as possible. Inorganic tropospheric chemistry considers ozone, NO, $NO_2$, $NO_3$, $N_2O_5$, HONO, $HNO_3$, $HNO_4$, HCN, CO, $H_2$, OH, $HO_2$, $H_2O_2$, $NH_3$, chlorine and bromine species, $SO_2$, and oxygen atoms. The complete mechanism of JAM002 (species and equations) can be found in the directory mecca/eqn/jam/ in the supplement. In total, JAM002 contains 246 species and 733 reactions, including 142 photolysis reactions. Detailed information can be found in Schultz et al. (2018).

### 2.2.4 MCM

The Master Chemical Mechanism (MCM) describes in detail the tropospheric degradation of more than a hundred VOCs (Jenkin et al., 1997; Saunders et al., 2003). It is widely used as the reference mechanism for modeling studies of atmospheric processes. Although the standard organic chemistry mechanism in MECCA (MOM, described above) is sufficient for many model applications, a more explicit mechanism can be necessary when studying specific VOCs. For example, the fate of limonene ($C_{10}H_{16}$) emitted from boreal forests is not included in the standard MECCA mechanism. To use the MCM reactions inside MECCA, the new tool xmcm2mecca has been added, which converts an extracted subset of MCM[1] to a KPP equation file that is compatible

---

[1]http://mcm.leeds.ac.uk/MCM

**Table 3.** Simplified example list of species with overall interaction coefficients (OICs). The full mechanism includes all species; the skeletal mechanisms s1, s2, and s3 only include species above a certain OIC threshold. Targets with OIC = 1 are always included. The color coding of the skeletal mechanism (used also in Fig. 4) shows in which mechanism a species occurs. For example, orange is used for species which are included in the full mechanism and in s1 but not in s2 and s3.

| species | OIC | full | s1 | s2 | s3 |
|---|---|---|---|---|---|
| N | 0.000000E+00 | 🔴 | | | |
| PERPINONIC | 1.944015E−04 | 🔴 | | | |
| PINENOL | 3.939767E−04 | 🔴 | | | |
| PINALNO3 | 5.772079E−04 | 🔴 | | | |
| PINONIC | 9.361802E−04 | 🟠 | 🟠 | | |
| APINAOO | 9.383650E−04 | 🟠 | 🟠 | | |
| APINBOO | 9.383650E−04 | 🟠 | 🟠 | | |
| PINALOOH | 1.033250E−03 | 🟡 | 🟡 | 🟡 | |
| BPINANO3 | 1.147639E−03 | 🟡 | 🟡 | 🟡 | |
| BPINAOOH | 1.260848E−03 | 🟡 | 🟡 | 🟡 | |
| MEK | 1.282217E−03 | 🟡 | 🟡 | 🟡 | |
| CAMPHENE | 1.473224E−03 | 🟢 | 🟢 | 🟢 | 🟢 |
| SABINENE | 2.525735E−03 | 🟢 | 🟢 | 🟢 | 🟢 |
| CARENE | 2.877949E−03 | 🟢 | 🟢 | 🟢 | 🟢 |
| APINENE | 6.029040E−03 | 🟢 | 🟢 | 🟢 | 🟢 |
| BPINENE | 9.412960E−03 | 🟢 | 🟢 | 🟢 | 🟢 |
| C5H8 | 2.914117E−01 | 🟢 | 🟢 | 🟢 | 🟢 |
| MVK | 3.432776E−01 | 🟢 | 🟢 | 🟢 | 🟢 |
| PAN | 3.505309E−01 | 🟢 | 🟢 | 🟢 | 🟢 |
| CH4 | 5.527123E−01 | 🟢 | 🟢 | 🟢 | 🟢 |
| NO2 | 9.998463E−01 | 🟢 | 🟢 | 🟢 | 🟢 |
| HCHO | 1.000000E+00 | 🟢 | 🟢 | 🟢 | 🟢 |
| HO2 | 1.000000E+00 | 🟢 | 🟢 | 🟢 | 🟢 |
| NO | 1.000000E+00 | 🟢 | 🟢 | 🟢 | 🟢 |
| O3 | 1.000000E+00 | 🟢 | 🟢 | 🟢 | 🟢 |
| OH | 1.000000E+00 | 🟢 | 🟢 | 🟢 | 🟢 |

with MECCA. The User Manual provides a detailed description of this new tool.

## 2.3 Mechanism intercomparison

Having all mechanisms implemented in the same modeling system enables mechanism intercomparison studies under exactly the same conditions. This approach ensures that any resulting differences come from the chemical mechanism, not from any other parts of the model. We have performed such an intercomparison for MOM, CB05BASCOE, MOZART, MIM1, JAM002 and a comparable subset of MCM. Details about these model runs and the results for all species are available in the testsuite/cams directory in the supplement. Some representative results are shown in Fig. 3. All mechanisms show a very similar decay of the initial isoprene because they all use similar rate constants for the main reactions of isoprene with ozone, OH and $NO_3$. In contrast, the results for the terpenes differ. In CB05BASCOE and MOZART, the rate constants for the lumped terpenes

are taken from $\alpha$-pinene. In the other mechanisms, $\beta$-pinene (and other terpenes) are considered individually. Since $\beta$-pinene reacts with ozone much slower than $\alpha$-pinene, the explicit treatment of $\beta$-pinene in the mechanism leads to a slower decay of the terpenes than in the lumped mechanisms. With respect to peroxyacetyl nitrate (PAN), especially CB05BASCOE shows very different values during the first day of the simulation. The calculated diurnal cycles of ozone, OH and $NO_2$ are similar for all mechanisms but their absolute values vary. Highest concentrations are produced by MIM1 and MCM, the lowest by CB05BASCOE and JAM002. MOM and MOZART are in between.

It is interesting to compare our results to a mechanism intercomparison study conducted about 10 years ago by Emmerson and Evans (2009), who partially used predecessors of the mechanisms in our code. They found significant differences for both PAN and isoprene. Using the present-day versions of the mechanisms, we still see differences for PAN but very similar results for isoprene.

## 2.4 Skeletal mechanism reduction

In the area of fuel combustion research, chemical models require highly complex mechanisms to describe ignition, flame propagation, and other properties. In order to save computing time, several methods have been developed to create a simplified chemical mechanism (called skeletal mechanism), which still produces similar results as the full mechanism (e.g., Tomlin and Turányi, 2013). One of these methods is DRGEP (Directed Relation Graph with Error Propagation), which was introduced by Pepiot-Desjardins and Pitsch (2008) and implemented into the MARS (Mechanism Automatic Reduction Software) model by Niemeyer et al. (2010) and Niemeyer and Sung (2011). The DRGEP code from MARS has been implemented in CAABA/MECCA, making the skeletal reduction method available for atmospheric chemistry mechanisms. The most important quantities of DRGEP are briefly explained below, full details can be found in Niemeyer et al. (2010).

**Targets:** Important chemical species, for which the skeletal mechanism has to produce similar results as the full mechanism.

**Sample points:** A set of environmental conditions (temperature, pressure, concentrations of chemical species) simulated by the chemistry model.

**Interaction coefficients (DIC, PIC, OIC):** The importance of chemical species in a mechanism is defined in terms of several interaction coefficients. The direct interaction coefficient (DIC) describes the importance of one species for another, based on its normalized contribution to production/consumption rates through reactions involving both species. Then, a graph search calculates a path interaction coefficient (PIC) based on the prod-

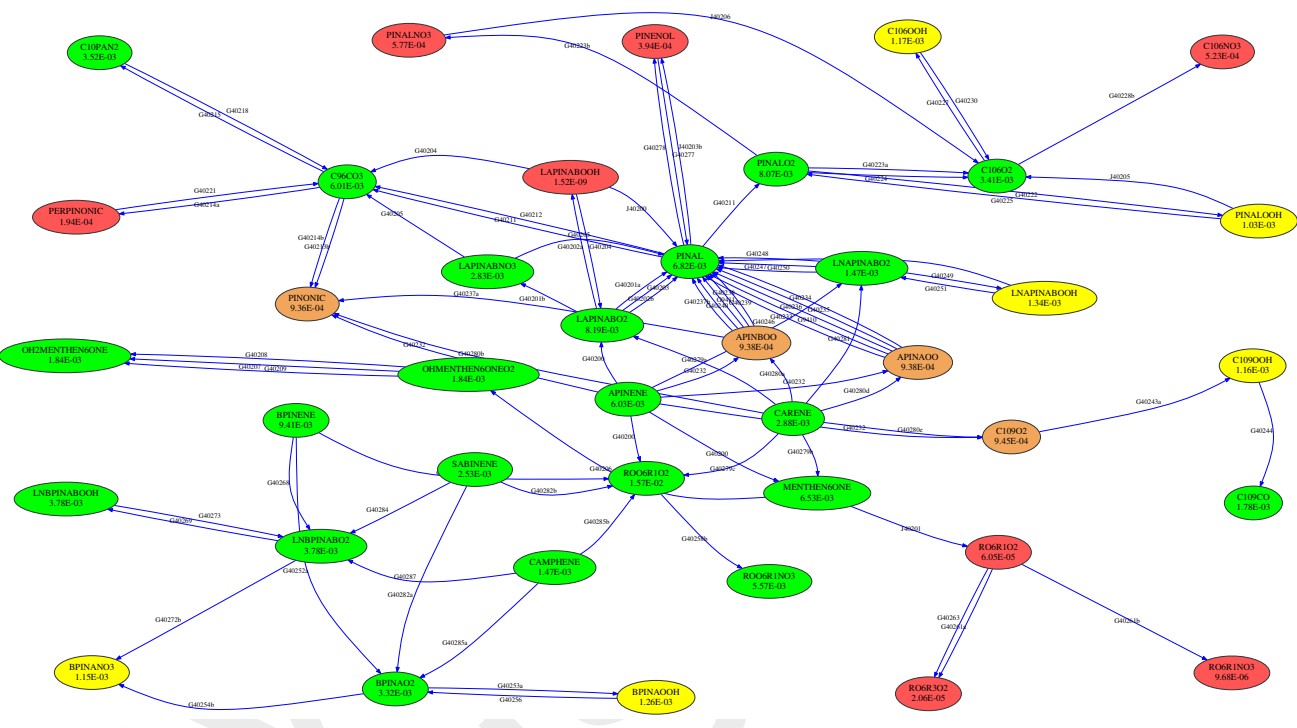

**Figure 4.** Skeletal reduction of terpene chemistry in the MOM reaction scheme (only $C_{10}$ species are shown here). Vertex colors and OIC values correspond to those in Tab. 3: Only the green and yellow species are kept in the reduced mechanism.

uct of direct interaction coefficients along the path from target to species, where nodes represent species and weighted directed edges represent DICs. Finally, the overall interaction coefficient (OIC) is the maximum of all PICs between target and species. It is calculated for all sample points and expressed as a value between 0 (unimportant) and 1 (important). For targets, OIC = 1 by definition. OIC values are only calculated for the full mechanism.

**Error $\delta_{\mathrm{skel}}$:** A normalized value describing the error when using a skeletal mechanism instead of the full mechanism. A skeletal mechanism is suitable if $\delta_{\mathrm{skel}} < 1$ for all targets and sample points. To allow individual weighting, the calculation of $\delta_{\mathrm{skel}}$ depends on a target threshold AbsTol and a maximum acceptable relative tolerance RelTol, which are defined for all targets:

$$\delta_{\mathrm{skel}} = \left| \frac{\max(x_{\mathrm{skel}}, \mathrm{AbsTol})}{\max(x_{\mathrm{full}}, \mathrm{AbsTol})} - 1 \right| / \mathrm{RelTol} \qquad (5)$$

where $x_{\mathrm{full}}$ and $x_{\mathrm{skel}}$ are the mixing ratios calculated with the full and the skeletal mechanism, respectively.

**OIC threshold $\varepsilon_{\mathrm{ep}}$:** A chemical species is considered important if OIC(species) $> \varepsilon_{\mathrm{ep}}$. The final $\varepsilon_{\mathrm{ep}}$ calculated

by DRGEP is the maximum value for which $\delta_{\mathrm{skel}} < 1$ still holds.

To test the skeletal mechanism generation, we chose HCHO, $HO_2$, NO, $O_3$, and OH as targets, allowing a relative tolerance of RelTol = 20 % for mixing ratios above a threshold of AbsTol = 1 pmol/mol. Sample points were extracted from a global atmospheric chemistry simulation with a setup similar to that presented by Lelieveld et al. (2016). The chemical compositions were taken from several boxes at two altitudes (at the surface and at about 1 km). As we want the skeletal mechanism to perform well not only at typical concentrations of the targets but also when they are very high or very low, we picked boxes where the targets reach their minimum, average, or maximum concentrations, respectively. This resulted in the generation of 30 sample points (5 targets times (min/ave/max) times 2 altitudes), covering a wide range of values. The full mechanism contained the complete set of species from MOM (Sect. 2.1). To illustrate the mechanism, the subset describing terpene chemistry is shown in Fig. 4. The importance (OIC values) of a few selected species is shown in Tab. 3. Three skeletal mechanisms (s1, s2, s3) were generated, reducing the number of species from 663 in the full mechanism to 462, 429, and 411, respectively. The number of reactions was reduced from 2091 to 1444, 1320, and 1262, respectively. The third skeletal mech-

anism (s3) was rejected because it did not fulfill the criterion $\delta_{\mathrm{skel}} < 1$. Results obtained with the full mechanism and with s2 were compared in a global simulation, as described below in Sect. 4.2.

## 2.5   Kinetic and isotope tagging

We have updated the sub-submodel MECCA-TAG (Gromov et al., 2010), which had been introduced in version 3.0 of CAABA. Several improvements to the kinetic tagging technique were implemented. These new features include:

- Selectable composition transfer mode: Depending on the research question, prescribed-, molecular- or element-weighted composition transfer may be selected. These modes determine the shares with which each reactant contributes to the products in the tagged chemical reactions: according to user-specified weightings, proportional to the reacting molecules count, or following the given element (e.g., C or H) content, respectively. Whilst the latter mode is intrinsic to isotope tagging, the others may be used for custom tagging configurations, e.g., product yield calculations.

- Diagnostics for unaccounted production or loss of elemental composition: MECCA-TAG optionally adds passive diagnostic species to the tagged reactions with unbalanced transfer of the element of interest. This helps to quantify the amount of atoms the chemical mechanism receives from or loses to "nothing", including the isotope composition of such mass-balance violations.

- The new "class shifting" tagging mode: This mode allows migration of molecules between the tagging classes in specified reactions, which allows quantifying various exchange processes in the mechanism. For instance, one can distinguish oxidation generations: in reactions with given oxidants the products become "promoted" to the tagging class of the next oxidation generation. Another application of "class shifting" is quantifying the efficiency of recycling chains. In essence, such is the "online" implementation of the approach similar to that of Lehmann (2004), with the number of tagging classes defining the maximum of the recycling sequences it is possible to follow.

The range of MECCA-TAG applications was extended with new tagging setups/configurations:

- Radiocarbon configurations, which facilitate simulating the $^{14}$C content in a desired set of species, including the routines for calculating abundances using conventional units like pMC (percent Modern Carbon).

- Hydrogen isotope chemistry: Now MECCA-TAG allows tracing pathways of H transfer between the species

in the mechanism. Furthermore, D/H isotope chemistry (including relevant kinetic isotope effects for $HO_x$ and $C_1 - C_2$ chemistry) are included. The configuration and calculations of the composition transfer were extended with the possibility to specify isotope branching ratios necessary for the consistent D/H kinetics simulations. Both H transfer and D/H chemistry are currently evaluated in stratospheric setups of CAABA (Frank et al., 2018).

- $O_2$ clumped isotope chemistry: simulation of non-stochastic distributions of $^{18}O^{18}O$ and $^{17}O^{18}O$ isotopologues ($\Delta_{36}$ and $\Delta_{35}$ signatures) resulting from $O(^3P)$-mediated temperature-dependent isotope exchange kinetics.

There are also some changes in the implementation and software requirements. There is no "doubling" mode anymore for evaluating the results of the optimized tagging. Performing kinetic tagging of the chemical mechanism with MECCA-TAG requires the Free Pascal Compiler (fpc[2], version $\geq 2.6$) at the time the xmecca script is run. The sub-submodel files are located in the mecca/tag/ directory of the distribution. The directory mecca/tag/cfg/ contains tagging configuration control files (*.cfg). The option to tag a newly created chemical mechanism is available in the xmecca script (also via batch files). Further details about the MECCA-TAG code development can be found in the file mecca/tag/CHANGELOG within the CAABA distribution.

## 3   Photolysis

CAABA contains several submodels which provide photolysis rate coefficients $j$, also called "$j$-values". The simple submodels READJ and SAPPHO have already been described by Sander et al. (2011a). READJ has not changed since version 3.0. SAPPHO photolysis rates can now be scaled using a common enhancement factor "efact" for all photolysis rates. This has for instance been used to simulate the very bright conditions within a cloud top (Heue et al., 2014). The updated and new photolysis submodels JVAL and RADJIMT are described in the sections below.

## 3.1   JVAL

The submodel JVAL inside the CAABA/MECCA model calculates $j$-values using the method of Landgraf and Crutzen (1998). It was first updated to the version described by Sander et al. (2014), and then additional changes were made. Many new photolysis reactions have been added, most of them related to either species from the MOM mechanism (CH3NO3, CH3O2NO2, CH3ONO, CH3O2, HCOOH, C2H5NO3, NOA, MEKNO3, BENZAL, HOC6H4NO2, CH3COCO2H,   IPRCHO2HCO,   C2H5CHO2HCO,

---

[2]https://www.freepascal.org/

**Table 4.** New upper atmosphere reactions for which RADJIMT provides $j$-values.

| | | | |
|---|---|---|---|
| $O(^3P)$ | $+ e^*$ | $\rightarrow$ | $O^+ + e^- + e^*$ |
| $O_2$ | $+ e^*$ | $\rightarrow$ | $O_2^+ + e^- + e^*$ |
| $O_2$ | $+ e^*$ | $\rightarrow$ | $O^+ + O(^3P) + e^- + e^*$ |
| $N_2$ | $+ e^*$ | $\rightarrow$ | $N_2^+ + e^- + e^*$ |
| $N_2$ | $+ e^*$ | $\rightarrow$ | $N^+ + N + e^- + e^*$ |
| $N_2$ | $+ e^*$ | $\rightarrow$ | $N^+ + N(^2D) + e^- + e^*$ |
| $N_2$ | $+ e^*$ | $\rightarrow$ | $N + N(^2D) + e^*$ |
| $O_2$ | $+ h\nu$ | $\rightarrow$ | $O(^3P) + O(^1D)$ |
| $O_2$ | $+ h\nu$ | $\rightarrow$ | $O_2^+ + e^-$ |
| $O_2$ | $+ h\nu$ | $\rightarrow$ | $O^+ + O(^3P) + e^-$ |
| $O(^3P)$ | $+ h\nu$ | $\rightarrow$ | $O^+ + e^-$ |
| $H_2O$ | $+ h\nu$ | $\rightarrow$ | $H_2 + O(^1D)$ |
| $N_2$ | $+ h\nu$ | $\rightarrow$ | $N_2^+ + e^-$ |
| $N_2$ | $+ h\nu$ | $\rightarrow$ | $N^+ + N + e^-$ |
| $N_2$ | $+ h\nu$ | $\rightarrow$ | $N^+ + N(^2D) + e^-$ |
| $N_2$ | $+ h\nu$ | $\rightarrow$ | $N + N(^2D)$ |
| $NO$ | $+ h\nu$ | $\rightarrow$ | $NO^+ + e^-$ |

C3H7CHO2HCO, PeDIONE24, PINAL2HCO) or organic halogen compounds (CF2ClCFCl2, CH3CFCl2, CF3CF2Cl, CF2ClCF2Cl, CHF2Cl, CHCl3, CH2Cl2). Besides, bugfixes were necessary regarding incorrect temperature dependencies of the ozone and OCS cross sections in the input data.

### 3.2 RADJIMT

RADJIMT is a new submodel that provides dissociation and ionization rates due to absorption of light and energetic photoelectrons in the mesosphere and thermosphere (see Tab. 4). It is part of the upper atmosphere extension of MESSy initially described by Baumgaertner et al. (2013), which was partly based on the implementations from the middle and upper atmosphere model CMAT2 (Harris, 2001; Dobbin, 2005; Dobbin and Aylward, 2008). For upper atmosphere simulations with CAABA, MECCA was extended by the relevant chemical species (electrons and ions) and reactions (labeled %Up in gas.eqn). For the respective literature sources, see meccanism.pdf in the supplement.

Photodissociation and photoionization due to the absorption of solar X-ray, EUV, and UV radiation are calculated using fluxes from the SOLAR2000 empirical model (Tobiska et al., 2000), the GLOW model (Solomon et al., 1988), as well as data presented by Henke et al. (1993) and Fennelly and Torr (1992). Relative partitioning between the possible products of the ionization process are based on the model of Strickland and Meier (1982) and Fuller-Rowell (1993).

For solar zenith angles larger than $75°$, the atmospheric column of each absorbing species is calculated using an approximation of the Chapman grazing incidence function (Smith and Smith, 1972).

Reaction enthalpies in kJ/mol (exothermic chemical heating) are provided as a product of the relevant chemical reactions when "set enthalpy=y" is defined in the MECCA batch file. Radiative heating and cooling is also calculated by the submodel (variable "heatrates").

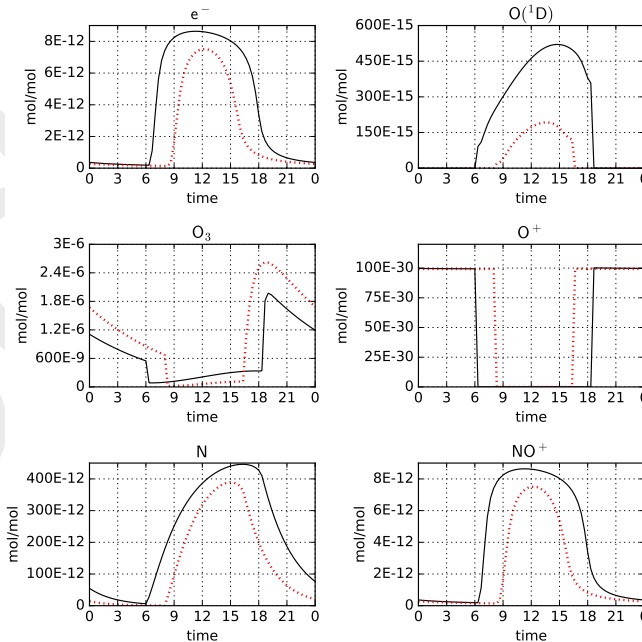

**Figure 5.** Model-calculated mixing ratios from an upper atmosphere simulation with MECCA and RADJIMT: Diurnal cycles for 4 January (after 3 days of spinup) for the equator (black) and a latitude of $50°$ N (red). Time is in hours, with local noon at 12. See Sect. 3.2 for further details.

As an example, we have performed simulations with CAABA using the MECCA and RADJIMT submodels. The mechanism was created using the batch file mtchem.bat, which selects reactions of the upper atmosphere labeled %Up. The model setup in caaba_mtchem.nml was used: The temperature was kept constant at 195 K, and the pressure was set to 0.5 Pa (approximately 85 km). The model starts on 1 January. Chemical species were initiliased using the values provided by Brasseur and Solomon (2005) in their Tables A.6.1 and A.6.2. The default timestep length of 20 minutes was used. For MECCA and RADJIMT, the default settings were used. Model-calculated mixing ratios for a few selected species are shown in Fig. 5. A comprehensive set of plots is available in radjimt_mixrat.pdf and radjimt_jvalues.pdf in the supplement.

### 3.3 DISSOC

The new MESSy submodel DISSOC is based on the photolysis scheme by Meier et al. (1982). Briefly, it calculates a table of the so-called enhancement factor, which is basically the ratio of the actinic flux at a specific location to the solar irradiance at the top of the atmosphere. The enhancement factor depends on the pressure level, solar zenith angle and

wavelength. Input data are the solar irradiance at the top of the atmosphere, absorption cross sections, ozone and oxygen profiles. For the implementation into global models, the input profiles are allowed to be latitude-dependent, which increases the dimensions of the enhancement factor table from three to four. Photolysis rates are calculated from the tabulated enhancement factor as a wavelength integral over the product with the absorption cross sections. The calculation is formulated in spherical geometry, such that it can be also applied to zenith angles above $90°$. Rayleigh scattering is calculated based on Nicolet et al. (1982). Absorption cross sections are taken from the current JPL recommendations (Burkholder et al., 2015).

The code was first implemented by Lary and Pyle (1991) and coupled to a stratospheric chemistry-box model (Müller et al., 1994). Becker et al. (2000) improved the treatment of the diffuse actinic flux and corrected an implementation error of Meier et al. (1982). The extension to the use of multiple latitudes was introduced within the development of the model CLaMS (McKenna et al., 2002). The possibility to calculate diurnally averaged photolysis rates was introduced for the simplified fast chemistry setup used in multi-annual CLaMS simulations (Pommrich et al., 2014).

In the current configuration, DISSOC determines the photolysis rates for 38 photolysis reactions that are primarily of relevance in stratopsheric chemistry. A standard setup contains 36 pressure levels, 18 latitude bins, and 28 solar zenith angle bins (of which 8 are above $90°$). Of the 203 standard wavelength intervals between $116\,\mathrm{nm}$ and $850\,\mathrm{nm}$, typically only the 159 intervals above $175\,\mathrm{nm}$ are used for tropospheric and stratospheric applications.

## 4  MECCA in the MESSy modeling system

Apart from using MECCA inside the CAABA box model, it is also possible to connect MECCA chemistry to a trajectory or global, 3-dimensional model via the MESSy infrastructure (Jöckel et al., 2010, 2016). Recent developments of MECCA shown in this section are related to its implementation inside MESSy.

### 4.1  TRAJECT

The TRAJECT submodel by Riede et al. (2009) allows simulations of atmospheric chemistry along pre-calculated Lagrangian trajectories. For this purpose, the air parcel simulated by CAABA is moved through space and time along a trajectory taken from an external input file, while simulating atmospheric photochemistry with MECCA and JVAL. More generally, TRAJECT allows to prescribe physical boundary conditions for CAABA box model simulations. A typical application is the simulation of atmospheric trajectories (balloon measurements or backward trajectories). However, laboratory conditions (e.g., in a flow reactor) can also be

prescribed. The previous TRAJECT version, described by Sander et al. (2011a), has been updated. The output is now more consistent with the trajectory input file, as physical information is now written out beginning with the first time step instead of the second. In general, an integration time step of chemical kinetics is always performed with the physical parameters given for the end of the time step. In that way, the mixing ratios written out at the end of a time step are consistent with the physical conditions at that point. Also, solar zenith angle and local time at the end of a time step are now consistent with the given longitude and latitude for that trajectory point.

In addition to the trajectory input file, an external input file with $j$-values for $NO_2$ can be used to scale all $j$-values with the factor:

$$\mathrm{jfac} = \frac{j(NO_2, \mathrm{external})}{j(NO_2, \mathrm{JVAL})} \tag{6}$$

To facilitate the analysis of the scaling impact, jfac is now written to output. Scaling thresholds have been implemented to prevent artifacts that would occur when $j(NO_2, \mathrm{JVAL})$ is very small and the calculation of jfac approaches a division by zero.

### 4.2  PolyMECCA/CHEMGLUE

In a standard global model simulation, the MESSy submodel MECCA contains one chemical mechanism that is used for all grid boxes. This ensures a consistent chemistry simulation from the surface to the upper atmosphere. However, in some cases, it may be preferable to allow different mechanisms in different boxes, e.g., terpene chemistry only in the troposphere and ion chemistry only in the mesosphere.

With the script xpolymecca, several independent chemical MECCA mechanisms can be produced. The first mechanism has the name "mecca", as usual. Additional mechanisms are labeled with a three-digit suffix. For example, the code of mechanism 2 is contained in messy_mecca002_kpp.f90 and related files.

To select an appropriate mechanism at each point in space and time, the MESSy submodel CHEMGLUE has been written. The name of the submodel was chosen because CHEMGLUE can also "glue" together different chemical mechanisms at the border where a chemical species is included in one mechanism but not in the other. CHEMGLUE defines the new channel object "meccanum", which contains the mechanism number for each grid point. These values can either be selected statically, e.g., depending on the model level number or the sea-land fraction mask. Alternatively, a dynamic (time-dependent) selection based on chemical or meteorological variables is possible, e.g., pressure, temperature, or the concentrations of ozone or isoprene.

Note that even when different boxes of a global model simulation use different chemistry mechanisms, the set of tracers contains all species from all mechanisms for all boxes.

The implementation ensures binary identical results when one chemical mechanism ("mecca") is replaced by two identical copies of it ("mecca" and "mecca002").

For a more realistic test, we created two different chemical mechanisms for organics. In the first mechanism, only the oxidation of methane is considered, and all non-methane hydrocarbons are neglected. The second (FULL) contains the full set of MOM (Sect. 2.1) reactions. CHEMGLUE selects the second mechanism whenever the mixing ratios of organics are above a threshold (isoprene $> 100$ pmol/mol, $\alpha$-pinene $> 100$ pmol/mol, or toluene $> 10$ pmol/mol). To investigate how much CPU time can be saved and how much the simplification affects the results, we have performed global test simulations based on the ECHAM5/MESSy atmospheric chemistry (EMAC) model by Jöckel et al. (2016). The horizontal resolution was T42 ($2.8° \times 2.8°$), with 47 vertical levels. Starting on 1 Jan 2009, one month was simulated. To facilitate the intercomparison between the simulations, the feedback of chemistry on the meteorology was switched off. Three different chemical scheme were tested:

1. FULL: Full MOM chemistry was activated throughout the atmosphere.

2. POLY: PolyMECCA/CHEMGLUE switches between the full MOM chemistry and the methane-only chemistry as described above.

3. SKEL: The skeletal mechanism s2 as described in Sect. 2.4 was activated throughout the atmosphere.

The CPU usage for the POLY and SKEL simulations are 62 % and 65 % of the FULL simulation, respectively. Results are shown in Fig. 6. Overall, the agreement between the simulations is quite good, considering that the simplified mechanisms neglect many reactions.

### 4.3 CHEMPROP

Chemical properties of the species in the reaction mechanism are needed at many locations in the model, e.g., molar mass ($M$), Henry's law constants ($H$), accommodation coefficients ($\alpha$), acidity constants ($K_A$), and ion charge number ($z$). These values have so far been stored at different locations in the code (gas.tex, messy_cmn_gasaq.f90, and elsewhere). Because maintaining data that are spread over several source files is tedious and error-prone, the new CHEMPROP database has been created, which stores all values centrally in the ASCII table messy_main_tracer_chemprop.tbl. MECCA (and other submodels) can access these chemical property data via MESSy tracer containers, as described by Jöckel et al. (2008).

### 5 Further changes

– The new subroutines dilute and dilute_once dilute the concentrations of chemicals in an air parcel by mixing it with unperturbed air. This can for example be used for modeling chemistry in an expanding volcanic or smog plume. An alternative usage for these subroutines is the simulation of the flow in and out of a reaction chamber (e.g., van Eijck et al., 2013).

– A new functionality has been implemented for the external initialization of chemical species from a netCDF file: If the time axis of the input file contains more than one point, the time values are used to interpolate mixing ratios at model start time. This is convenient for bundling several initializations into one file, for instance to initialize several CAABA simulations from different points along a trajectory with recorded mixing ratios (see also Sect. 4.1). If the time axis of the input file contains only one point, the mixing ratios are read into CAABA, regardless of the time value.

– We extended CAABA with parameters to optionally control the output step frequency (output_step_freq) and the output synchronization frequency (output_sync_freq). The first variable sets the frequency at which values are written to the output. A value of output_step_freq $= \alpha$ skips $\alpha - 1$ timesteps and writes only every $\alpha$-th time step to the output file. The second variable controls the output synchronization. Data are buffered for output_sync_freq time steps before they are written to the output files. Both parameters enable the user to carry out very long box model simulations without being constrained by machine I/O performance, and they can individually regulate the output file size. A high value of output_sync_freq has a positive effect on performance. However, in case of machine failure buffered output steps are lost.

– The treatment of humidity has been improved. Now specific as well as relative humidity (RH) are available throughout CAABA, and can be interconverted with generic conversion functions. Of the two, specific humidity is the more robust variable for humidity because the definition of RH can be based on either partial pressure or on specific humidity (Jacobson, 1999). There are various parameterizations for saturation water vapor pressure, and RH can be defined over liquid surface even below 0 °C, if supercooling is allowed. Functions that use humidity as input (concentration of air, conversion between humidity and water vapor concentration) now use the unambiguous specific humidity. If necessary, it is derived from relative humidity taking all of the above considerations into account.

– For better model time control, two boolean namelist parameters have been introduced: l_groundhogday=T repeats a diurnal cycle while l_freezetime=T repeats a certain point in time, effectively freezing the solar zenith angle.

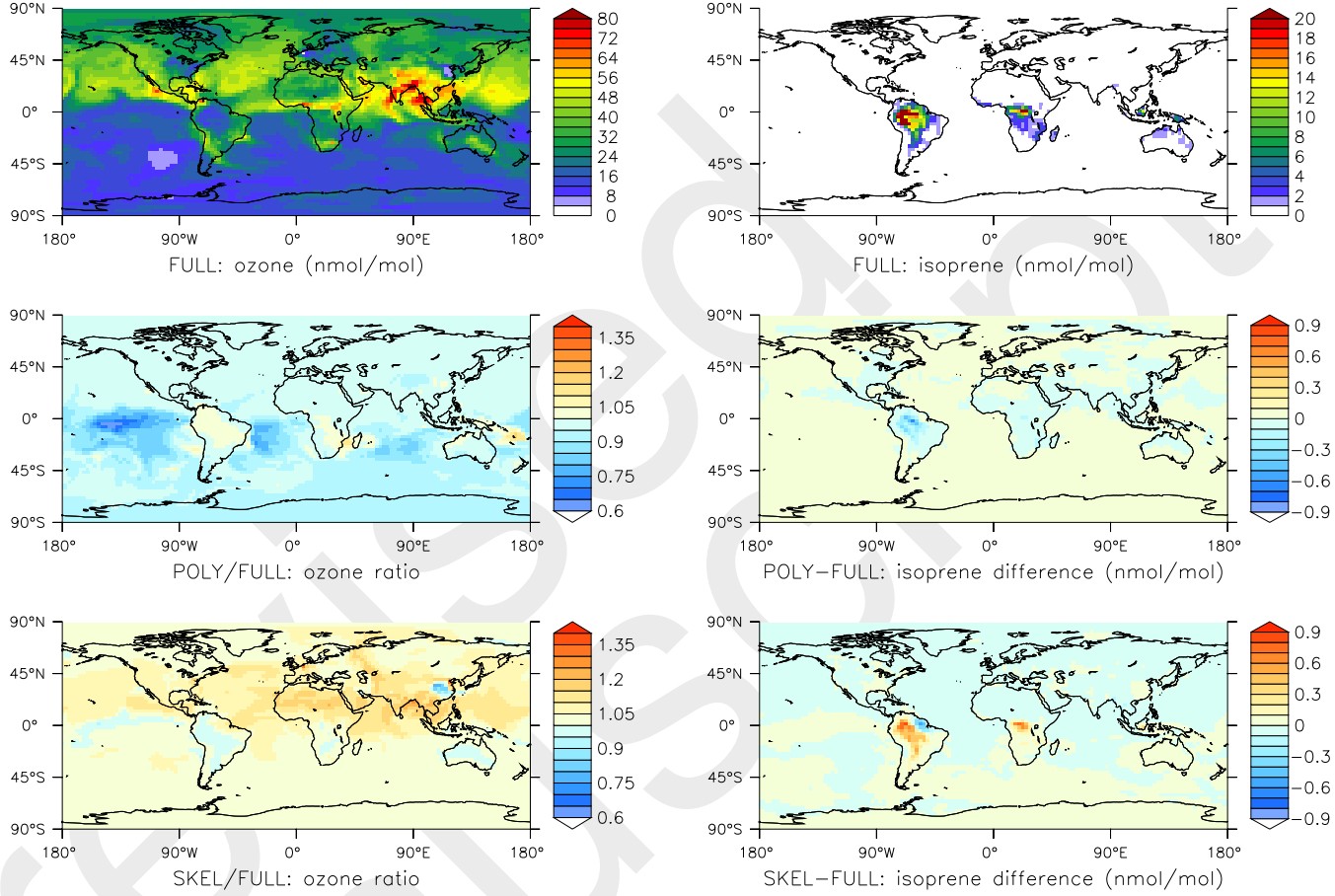

**Figure 6.** Results of the global comparison between the FULL, POLY, and SKEL mechanisms (see Sect. 4.2 for details). Shown are suface mixing ratios of ozone (left column) and isoprene (right column) at the end of the simulation, i.e. after one month. The top row shows results obtained with the FULL chemistry mechanism. The middle row compares POLY to FULL, and the bottom row compares SKEL to FULL.

– The selection of various chemical species to define steady state has been simplified to allow for more flexibility in the criteria. The progress towards the defined steady state is now logged during CAABA runtime. Artifacts by species' concentrations close to zero are now prevented.

– Several shell scripts have been converted to python (xcaaba.py, multirun.py, montecarlo.py). They use the netcdf4 interface and don't depend on the availability of the NetCDF operators (ncks etc.) anymore. Currently, the python scripts are in beta-testing. In future versions, they will replace the current tcsh scripts.

– Model results can now be visualized with the python script caabaplot.py using matplotlib. The previously used ferret scripts are still included but not actively supported anymore.

– Complex reaction mechanism can be interpreted as graphs, with species representing vertices and reac-

tions representing edges. To visualize and analyze these graphs, the graph-tool software by Peixoto (2014) can now be used. For example, Fig. 4 was created with graph-tool.

– Rate coefficients have been updated to the latest JPL recommendations (Burkholder et al., 2015) and recent laboratory studies. A complete list of chemical reactions, rate coefficients, and references is available in the supplement (meccanism.pdf).

– The kinetic preprocessor KPP (Sandu and Sander, 2006) performs the numerical integration of the chemical reaction mechanism. It has been updated to the latest version 2.2.3, which contains a number of small fixes throughout the code[3].

– The scripts check_eqntags.py and check_eqns.pl check the internal consistency of the chemical mechanism.

---

[3]http://people.cs.vt.edu/~asandu/Software/Kpp

– Details of all new features have been added to the updated User Manual, which now also includes an index. Additional minor bug fixes can be found in the CHANGELOG file.

## 6 Summary and outlook

We have presented the current version of the atmospheric chemistry module MECCA-4.0, which includes several new features: Skeletal mechanism reduction, the MOM chemical mechanism for organic compounds, optional inclusion of reactions from MCM and other chemical mechanisms, updated isotope tagging, and improved and new photolysis modules. When MECCA is connected to a global model, PolyMECCA and CHEMGLUE allow coexisting multiple chemistry mechanisms. CAABA/MECCA is now available to the research community.

Based on the model development described in this paper, our current and upcoming goals are (for work in progress, initials of the principal investigators are shown in parentheses):

– Reduce complex mechanisms to a size suitable for global model simulations (RS, KN).

– Perform a chemistry module intercomparison including CB05BASCOE and MOZART within a global chemistry modeling framework (Huijnen et al., 2019).

– Evaluate MOM chemistry and its effect on secondary aerosol formation (AP).

– Compare MOM chemistry to measurements obtained during the recent AQABA field campaign (HH).

– Advance our understanding of the role of organic compounds on the tropospheric $O_x$ and $HO_x$ budgets (DT).

– Compare model results with studies at the SAPHIR chamber (DT).

– Investigate the multiphase chemical pathways leading to organic acids and aerosols (DT).

– Simulate stratospheric isotope H exchanges between $CH_4$ and $H_2O$ (SG).

– Implement additional photolysis modules (e.g., CLOUDJ, TUV) and compare the resulting $j$-values (HH).

– Parallelize to distribute independent (e.g., Monte-Carlo or sensitivity) box model simulations on multiple cores (HH).

– Study the impact of aromatic compounds on atmospheric chemistry (RS, manuscript in preparation).

## Code and data availability

The CAABA/MECCA model code is available as a community model published under the GNU General Public License[4]. The model code can be found in the electronic supplement. In addition to the complete code, a list of chemical reactions, including rate coefficients and references (meccanism.pdf), and a User Manual (caaba_mecca_manual.pdf) are available in the manual/ directory of the supplement. For further information and updates, the MECCA web page at http://www.mecca.messy-interface.org can be consulted.

## Author contributions

RS develops and maintains the CAABA/MECCA software. AB provided RADJIMT. DC provided the aromatic chemistry mechanism of MOM. FF added code to control the model output. JUG provided DISSOC and helped with its implementation in MESSy. SG provided the MECCA-TAG sub-submodel. HH and ST provided code for the inclusion of the MCM reaction schemes. PJ contributed to several model development projects (MESSy modeling system, PolyMECCA, CHEMPROP, CHEMGLUE) and maintains the interfaces to ensure that the modules are not only compatible with the box model but also with the 3D models. VH contributed through initiating the provision of CAMS chemistry models for inclusion in MECCA, and for generation of the CB05BASCOE merged chemical mechanism. VK integrated CB05BASCOE and MOZART into MECCA. KN provided code for the skeletal mechanism generation. AP contributed to several model development projects (MESSy modeling system, scenarios for skeletal mechanism generation, MOM, CAMS, PolyMECCA testing). HR provided an update of the TRAJECT submodel. MS provided JAM002, and DT provided MOM.

**Supplementary material related to this article is available online at: https://doi.org/10.5194/gmd-0-1-2019-supplement.**

*Acknowledgements.* We thank Simon Chabrillat and Idir Bouarar for provision of the mechanisms BASCOE and MOZART, respectively. We also thank Tim Butler for contributing the diagnostic tool check_eqns.pl. Duy Cai added some photolysis reactions to JVAL. Tilo Fytterer and Stefan Versick discovered and reported the temperature dependence bug of the ozone and OCS photolyses in JVAL. VH acknowledges funding from the Copernicus Atmosphere Monitoring Service (CAMS).

_______________

[4]http://www.gnu.org/copyleft/gpl.html

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
