# Peer review of "The atmospheric chemistry box model CAABA/MECCA-4.0gmdd"

_Geoscientific Model Development, 2018_

## Referee Comment (RC1) · Anonymous Referee #1 · 6 Nov 2018

This paper accompanies the preliminary community release of version 4.0 of the CAABA/MECCA atmospheric chemistry box model into the GMD discussions forum, with the model code included as an electronic supplement. The paper summarises updates to the model, implemented since those described previously by Sander et al. (2005, 2011a). It first gives an overview of updates to the default chemical mechanism (MOM), and the provision of other optional mechanisms (CB05BASCOE, MOZART, JAM002 and MCM subsets) in compatible format. It goes on to outline other new features, which include skeletal mechanism reduction based on the DRGEP method, updated isotope tagging and improvements to the photolysis code, including the availability of new modules (JVAL, RADJIMT, DISSOC). Finally, a summary of recent developments of MECCA are given, that are related to its implementation into the MESSy

modelling system.

This paper acts as a reference document for both users and developers of this community model, and the publication of overview papers of this type is therefore important and necessary. In addition to providing a suitable citation that can acknowledge and reflect the considerable time and effort that goes into developing and maintaining a state-of-the-science code of this type, it acts as a record of how the representation of scientific knowledge evolves.

However, it therefore encounters the main difficulties and limitations commonly associated with such papers in that (i) they often provide only overview statements of the (many) methods, with limited justification in some areas, to ensure that the paper does not become too long and unwieldy; (ii) process representations are inevitably a snapshot of understanding at a given time, and therefore can lag behind the latest developments in fast moving subject areas; and (iii) they generally present few or no results to illustrate the performance of the methods and tools, these being deferred to future publications where they can be presented and discussed in greater detail. These inevitable limitations therefore provide some difficulties for reviewers when judging a paper of this type against some of the GMD review criteria.

Although a lot of useful information is presented, this paper suffers from some limitations in all the identified areas, as highlighted in the comments below. The authors should therefore consider whether they can provide more information and justification on some topics. Similarly, some illustration of the impacts of the updates (where relevant) might be useful. In practice, the simultaneous (or at least imminent) publication of an application paper might have been helpful. Several (presumably) ongoing and proposed activities are listed in Sect. 6, but there are no references to papers by the developers that are in press or in preparation.

Specific comments

Page 2, line 3: MESSy (Modular Earth Submodel System) should be defined.

Page 2, line 6: Non-methane hydrocarbon (NMHC) chemistry is listed as treated, but the term VOC is used everywhere else.

Page 2, line 25: "The rate of 1,4-H-shift for the MACRO2 radical is treated as predicted by Taraborrelli et al. (2012), which is about an order of magnitude lower than proposed by Crounse et al. (2012)."

However, the Crounse et al. (2012) study includes an experimental determination and is not simply "proposed" based on theory. Do the uncertainties in the theoretical value of Taraborrelli et al. (2012) encompass the experimental value of Crounse et al. (2012)? If so, surely the Crounse value should be applied. If not, further justification for the use of the Taraborrelli value is required.

In addition to this, I could not find any information on this specific reaction in Taraborrelli et al. (2012). Is that reference correct? It seems that Taraborrelli et al. (2012) considers 1,5 H-shifts involving transfer of a hydroxyl H atom and 1,6 H shifts involving transfer from CH2OH groups (focused on OH-isoprene-O2 radicals), whereas Crounse et al. (2012) considers the 1,4 H-shift involving transfer of the formyl H atom in MACRO2. Are the rates of two different types of H-shift reaction therefore being compared?

Page 4, section 2.1.1. This section presents a description of how the treatment of reactions of OH with VOCs (and their degradation products) has been updated, e.g. the use of the Peeters et al. (2007) approach for the reactions with alkenes. Some additional clarification of the methods would be helpful, including the following:

i) Although a very useful reference data set, the Atkinson et al. (2006) IUPAC evaluation is now quite old. Updates, refinements and expansions to the IUPAC evaluation are available at http://iupac.pole-ether.fr/. Although some of the preferred values may be unchanged from Atkinson et al. (2006), it seems strange not to take advantage of the more recent information.

ii) For the updates to the Kwok and Atkinson SAR method, what set of preferred data is

being used? Kwok and Atkinson used a much larger dataset than covered by Atkinson et al. (2006), which (with some exceptions) has a cut-off at C3.

iii) On page 4, line 9, it is stated that the substituent factors are "…updated or calculated ex novo by computing the relative rate coefficient of OH with the simplest VOC bearing the substituent relative to the one of its parent compound." First of all, it is not clear why the substituent factors are based on such a restricted dataset. Secondly, it is not at all clear what this means. The immediate suggestion is the substituent factor for an -OH group (for example) is determined from a comparison of the relative rate coefficient for CH3OH and CH4 (which is about 140 at 298 K). However, this is not compatible with the description of the method further down, which suggests the factor is probably actually determined from the rate coefficient for CH3OH, in conjunction with the Kwok and Atkinson value of kp (and kabst(-OH)). This latter procedure would give an F(-OH) value of 5.6 at 298 K, which is substantially greater than the Kwok and Atkinson value of 3.5, based on optimization to the full dataset of OH-containing compounds. If this is the revised method, I cannot see that this is any improvement on Kwok and Atkinson, and is almost certainly a retrograde step. Given that the current MCM uses the Kwok and Atkinson method, this would also not support the statement on page 4, line 11, "No rigorous evaluation of the SAR has been conducted and the estimation uncertainty is expected to be in the same range as for the SAR used by the MCM", which surely needs some further justification.

iv) There are a few other mentions of how the method adopts, or differs from, that used with the MCM. On page 4, lines 6 and 7 it states that "For the C6 to C11 species, the MCM rate coefficients are retained." and that they "….have no temperature-dependence and are only given at 298 K." However, inspection of the MCM website (http://mcm.leeds.ac.uk/MCM/) reveals that many OH rate coefficients for C6-C11 species are temperature dependent (e.g. those for C6-C10 n-alkanes). Does the given statement therefore mean that MOM is using temperature independent rate coefficients derived from the temperature-dependent expressions used in the MCM? If so, this

should be made clearer. It should also be noted that the MCM and GECKO-A teams recently published a comprehensive update to the way OH rate coefficients are to be calculated in the future (Jenkin et al., ACP, 2018), although it is recognised that this might be too recent for uptake into MOM, and is not yet used in MCM.

Page 5, line 1: should CH3CH2O2 be CH3CH2ONO2?

Page 8, line 8: Figure 3 is introduced here. Although this illustrates the comparative performance of the three mechanisms for the given scenario, no further discussion of the differences is given. Is the trace for CB05BASCOE obscured in the terpene panel, or are terpenes not represented?

Page 10, line 25: The term "Targets" is defined within the description of the skeletal reduction method. However, elsewhere they seem to be referred to as either "targets" or "target species".

---

## Referee Comment (RC2) · Anonymous Referee #2 · 27 Nov 2018

This paper by Sander et al., essentially provides a basic reference for updates and developments included in the update of the CAABA/MECCA atmospheric chemistry box modelling toolkit to version "4.0gmdd". This includes (1) updates and extensions to the chemical mechanism (Mainz Organic Mechanism), including updated SARs and photolysis rates; (2) provision of a range of alternative chemical mechanisms; (3) provision of a chemical skeletal reduction tool; (4) provision of a range of submodels for the calculation of photolysis rates throughout the atmosphere and (5) provision of a range of tools in order to connect MECCA to other modelling frameworks, including the MESSy 3-D global model. A range of other minor changes are also listed. The complete updated code discussed is made available to the community through the electronic supplementary along with a useful user manual section.

Although this type of is important and extremely useful to the community as a reference and "one stop shop" for this version of the CAABA-MECCA model, I found the paper reads very much like a report, listing the work done, and is therefore very dry to read. The descriptions of the work are basic and not particularly informative as to what has been done and why, and the impact of the changes. At the moment, the paper is only really useful to an experienced user of the CABBA/MECCA modelling tool kit.

It would have been good to see some impact of the updates carried out in terms of an evaluation of the chemical mechanistic updates against previous versions of MECCA/MOM as well as some evaluation against other chemical mechanisms, such as the more detailed benchmark MCM, over a range of atmospherically important chemical conditions. I realise that this is listed (and it is just a bullet pointed list) in the "Summary and outlook" section of the paper, but it is not clear if this work has been carried out, or is just a goal at the moment. In any case, some evaluation of the changes implemented in this version of CABBA/MECCA should be included in this paper before it is published in GMD.

More specific comments are given below:

- Perhaps mention "community model" in the title? (multi-purpose community atmospheric chemistry box model..)
- Introduction: A short descriptive definition of CABBA/MECCA and its history would be useful
- 2.1 Mainz Organic Mechanism (MOM) as the title
- Page 2, Line 14 – "explicit with a low degree of lumping" There for the scheme is not explicit!! More explanation needed here with respect to "lumping".
- Page 2, Line 16 – The latest version of the MCM is v3.3.1, which is focused around a detailed update/evaluation to the MCM isoprene chemistry (Jenkin et al., Atmos. Chem. Phys., 15, 11433–11459, 2015). MIM2 is based on MCMv3.1. Therefore, it would be useful to see how the updated MECCA

model chemistry compares to MCMv3.3.1 in terms of isoprene chemistry under a range of representative conditions.

- Page 2, Line 25 - Given the last comment, what is the justification of not using the experimentally derived Crouse et al., (2012) 1,4H-shift kinetic data?
- Page 2, Line 31 – Updated aromatic chemistry. Has any evaluation of this updated chemistry been carried out against chamber data? How does it compare to other model aromatic atmospheric chemical mechanisms (the prediction of NO to NO2 conversion and hence photochemical ozone formation, and chemical reactivity in various aromatics systems can vary significantly between different model chemistries.
- Figure 1. Please list species by category i.e. alkanes, alkenes, aromatics, oxygenates etc…
- Page 4**. 2.1.1. VOC reactions with OH** – This section is severally lacking in detail with respect to the updates actually carried out and the impact of such updates. Specifically:
    - When were the IUPAC rate constants last updated? Referencing the IUPAC website datasheets as Atkinson et al., (2006) is somewhat out of date.
    - The updated substituent factors form the original SAR of Kwok and Atkinson need to be shown here (in a table) so that they can be compared/contrasted to other SAR approaches and so the methodology is transparent to the community.
    - "No rigorous evaluation of the SAR has been conducted and estimation uncertainty is expected to be in the same range as the MCM" I strongly recommend that an evaluation of the updated SAR is carried out and compared to the original and other SAR approaches used in other models/chemical mechanisms.
- Page 4. "2.1.2 RO2 reaction with NOx" – Should be "with NOxy" as you also look at RO2 + NO3.
- Page 7. 2.1.4 RO2 permutation reactions, Line 11 – "The rate expressions are not from the MCM, except…" The rate expressions "not from the MCM" need to be defined and put into a Table here.
- Page 7. 2.1.5 Photo-induced reactions – Line 15 (and through-out) define photolysis rates as "$j$-values", i.e. $j(NO_2)$, and not "$J$-values" throughout.
- Define "HPALD and "PACALD"
- Figure 3 appears before it is mentioned in the text
- Page 8. 2.2 Other chemical mechanisms – some discussions, even brief ones about differences in the chemical schemes, would be useful
- Again, some evidence of evaluation of the updated MECCA/MOM chemistry against older versions and other detailed chemical mechanisms (such as the MCM) is needed here.
- Page 11, Table 1. Needs much more explanation here. What is the colour coding in the table?
- Page 10, line 11 – "from a global atmospheric chemistry simulation based on…", describe the scenarios used in this simulation
- Page 15, Line 13 – Figure 5 appears after the References?!
- Page 16. 4 MECCA in the MESSy modelling system – MESSy needs defining
- Page 20. Summary and outlook. Not actually a "summary" but a list of high level desirable goals. This should be re-written.

---

## Author Comment (AC1) · 26 Jan 2019

**Replies to reviewer 1**

> However, it therefore encounters the main difficulties and
> limitations commonly associated with such papers in that
> (i) they often provide only overview statements of the
> (many) methods, with limited justification in some areas,
> to ensure that the paper does not become too long and
> unwieldy; (ii) process representations are inevitably a
> snapshot of understanding at a given time, and therefore
> can lag behind the latest developments in fast moving
> subject areas; and (iii) they generally present few or no
> results to illustrate the performance of the methods and
> tools, these being deferred to future publications where
> they can be presented and discussed in greater detail.
> These inevitable limitations therefore provide some
> difficulties for reviewers when judging a paper of this
> type against some of the GMD review criteria. Although a
> lot of useful information is presented, this paper suffers
> from some limitations in all the identified areas, as
> highlighted in the comments below.
> The authors should therefore consider whether they can
> provide more information and justification on some topics.

We agree that these limitations exist to some extent and have tried to follow
the reviewer's specific suggestions as listed below.

> Similarly, some illustration of the impacts of the updates
> (where relevant) might be useful. In practice, the
> simultaneous (or at least imminent) publication of an
> application paper might have been helpful. Several
> (presumably) ongoing and proposed activities are listed in
> Sect. 6, but there are no references to papers by the
> developers that are in press or in preparation.

As requested, we have added more information in the outlook section, showing
the principal investigators of the work in progress and manuscripts in prepara-
tion.

> Page 2, line 3: MESSy (Modular Earth Submodel System)
> should be defined.

Done.

> Page 2, line 6: Non-methane hydrocarbon (NMHC) chemistry
> is listed as treated, but the term VOC is used everywhere
> else.

We have now added the definition VOCs = CH4 + NMHCs to the text.

```
> Page 2, line 25: "The rate of 1,4-H-shift for the MACRO2
> radical is treated as predicted by Taraborrelli et al.
> (2012), which is about an order of magnitude lower than
> proposed by Crounse et al. (2012)." However, the Crounse
> et al. (2012) study includes an experimental determination
> and is not simply "proposed" based on theory. Do the
> uncertainties in the theoretical value of Taraborrelli et
> al. (2012) encompass the experimental value of Crounse et
> al. (2012)? If so, surely the Crounse value should be
> applied. If not, further justification for the use of the
> Taraborrelli value is required. In addition to this, I
> could not find any information on this specific reaction
> in Taraborrelli et al. (2012). Is that reference correct?
> It seems that Taraborrelli et al. (2012) considers 1,5
> H-shifts involving transfer of a hydroxyl H atom and 1,6 H
> shifts involving transfer from CH2OH groups (focused on
> OH-isoprene-O2 radicals), whereas Crounse et al. (2012)
> considers the 1,4 H-shift involving transfer of the formyl
> H atom in MACRO2. Are the rates of two different types of
> H-shift reaction therefore being compared?
```

We thank the referees for spotting this issue and we apologize for it. Taraborrelli et al. (2012) did not consider 1,4-H shifts but after publication, very high-level quantum calculations of corresponding transition state were performed by L. Vereecken (personal communication with DT, 2013).

They resulted in a predicted energy barrier of 20.39 kcal/mol in contrast to the 19.0 kcal/mol reported by Crounse et al. (2012). We actually use the rate expression by Crounse et al. (2012) with an activation energy 1.39 kcal/mol higher. This results in a first-order decomposition rate constant of 0.04 s$^{-1}$, still much larger than usual atmospheric RO$_2$ sinks. We have now decided to adopt the rate constant by Crounse et al. (2012). Accordingly, we have modified the sentence

"The rate of 1,4-H-shift for the MACRO2 radical is treated as predicted by Taraborrelli et al. (2012), which is about an order of magnitude lower than proposed by Crounse et al. (2012)."

to

"The rate of the 1,4-H-shift for the MACRO2 radical is now calculated using the expression reported by Crounse et al. (2012)."

```
> Page 4, section 2.1.1. This section presents a description
> of how the treatment of reactions of OH with VOCs (and
> their degradation products) has been updated, e.g. the use
```

```
> of the Peeters et al. (2007) approach for the reactions
> with alkenes. Some additional clarification of the methods
> would be helpful
```

We have added a table (Table 1) showing the details on the origin of the rate constants and substituents factors that are used.

```
> i) Although a very useful reference data set, the Atkinson
> et al. (2006) IUPAC evaluation is now quite old. Updates,
> refinements and expansions to the IUPAC evaluation are
> available at http://iupac.pole-ether.fr/. Although some of
> the preferred values may be unchanged from Atkinson et al.
> (2006), it seems strange not to take advantage of the more
> recent information.
```

Although we neglected to mention this in the manuscript, we actually do consider the IUPAC udates published at `http://iupac.pole-ether.fr`. This information has now been added to the revised manuscript. References for individual reactions can be found in the file meccanism.pdf in the supplement.

```
> ii) For the updates to the Kwok and Atkinson SAR method,
> what set of preferred data is being used? Kwok and
> Atkinson used a much larger dataset than covered by
> Atkinson et al. (2006), which (with some exceptions) has a
> cut-off at C3.
```

Please see the new Table 1 for details.

```
> iii) On page 4, line 9, it is stated that the substituent
> factors are "... updated or calculated ex novo by
> computing the relative rate coefficient of OH with the
> simplest VOC bearing the substituent relative to the one
> of its parent compound." First of all, it is not clear why
> the substituent factors are based on such a restricted
> dataset. Secondly, it is not at all clear what this means.
> The immediate suggestion is the substituent factor for an
> -OH group (for example) is determined from a comparison of
> the relative rate coefficient for CH3OH and CH4 (which is
> about 140 at 298 K). However, this is not compatible with
> the description of the method further down, which suggests
> the factor is probably actually determined from the rate
> coefficient for CH3OH, in conjunction with the Kwok and
> Atkinson value of kp (and kabst(-OH)). This latter
> procedure would give an F(-OH) value of 5.6 at 298 K,
> which is substantially greater than the Kwok and Atkinson
```

> value of 3.5, based on optimization to the full dataset of
> OH-containing compounds. If this is the revised method, I
> cannot see that this is any improvement on Kwok and
> Atkinson, and is almost certainly a retrograde step. Given
> that the current MCM uses the Kwok and Atkinson method,
> this would also not support the statement on page 4, line
> 11, "No rigorous evaluation of the SAR has been conducted
> and the estimation uncertainty is expected to be in the
> same range as for the SAR used by the MCM", which surely
> needs some further justification.

We hope that the new Table 1 answers most of these questions. With respect to the quoted sentence, we acknowledge the inaccurate expression mentioning the parent compound. Neither Atkinson (1987) nor Kwok and Atkinson (1995) derived $F(-OH)$ as above but also did not detail how the values 3.4 and 3.5, respectively, were derived. In the last column of our new Table 1, we show how the substituent factor was calculated if it was not adopted from Kwok and Atkinson (1995). Our calculation results in $F^{sec}(-OH) = 3.44$ which is similar to the value from Kwok and Atkinson (1995).

> iv) There are a few other mentions of how the method
> adopts, or differs from, that used with the MCM. On page
> 4, lines 6 and 7 it states that "For the C6 to C11
> species, the MCM rate coefficients are retained." and that
> they "...have no temperature dependence and are only given
> at 298 K." However, inspection of the MCM website
> (http://mcm.leeds.ac.uk/MCM/) reveals that many OH rate
> coefficients for C6-C11 species are temperature dependent
> (e.g. those for C6-C10 n-alkanes). Does the given
> statement therefore mean that MOM is using temperature
> independent rate coefficients derived from the
> temperature-dependent expressions used in the MCM? If so,
> this should be made clearer. It should also be noted that
> the MCM and GECKO-A teams recently published a
> comprehensive update to the way OH rate coefficients are
> to be calculated in the future (Jenkin et al., ACP, 2018),
> although it is recognised that this might be too recent
> for uptake into MOM, and is not yet used in MCM.

We apologize for the confusion. The MCM rate constants for the closed-shell $C_6$ to $C_{11}$ species are retained. They are mostly constant values (SAR-estimated) but when experimental data is available, a temperature dependence is adopted, e.g. for the toluene + OH reaction. We have corrected the manuscript accordingly replacing

"For the $C_6$ to $C_{11}$ species, the MCM rate coefficients are retained. It is worth noting that the latter have no temperature-dependence and are only given at 298 K."

with

"For the $C_6$ to $C_{11}$ closed-shell species, the MCM rate coefficients are retained. It is worth noting that the SAR-estimated ones have no temperature-dependence and are only given at 298 K."

> Page 5, line 1: should CH3CH2O2 be CH3CH2ONO2?

We thank the reviewer for spotting this typo. We have changed the text accordingly.

> Page 8, line 8: Figure 3 is introduced here. Although this
> illustrates the comparative performance of the three
> mechanisms for the given scenario, no further discussion
> of the differences is given.

We have added more information about the model runs presented in Fig. 3. See also our replies to reviewer 2 regarding Fig. 3.

> Is the trace for CB05BASCOE obscured in the
> terpene panel, or are terpenes not represented?

The red line for the terpenes is indeed difficult to see in Fig. 3, it is mostly hidden by the green line. Both MOZART and CB05BASCOE contain similar chemical reactions for lumped terpenes, therefore the results are almost identical.

> Page 10, line 25: The term "Targets" is defined within the
> description of the skeletal reduction method. However,
> elsewhere they seem to be referred to as either "targets"
> or "target species".

Sorry for the confusion. "Targets" and "target species" are synonyms. For consistency, we now only use the term "targets" in the revised text.

**Replies to reviewer 2**

> Although this type of is important and extremely useful to
> the community as a reference and "one stop shop" for this
> version of the CAABA-MECCA model, I found the paper reads
> very much like a report, listing the work done, and is
> therefore very dry to read. The descriptions of the work
> are basic and not particularly informative as to what has
> been done and why, and the impact of the changes. At the
> moment, the paper is only really useful to an experienced
> user of the CABBA/MECCA modelling tool kit.

We are surprised that the reviewer finds the paper only useful for experienced users. The supplement contains the complete CAABA/MECCA model code as well as a very detailed User Manual. Every reader who is interested in it, can execute the model on their own computer. If there is anything else we can do to improve the accessibility and reproducibility of CAABA/MECCA, we're open for specific suggestions.

```
> It would have been good to see some impact of the updates
> carried out in terms of an evaluation of the chemical
> mechanistic updates against previous versions of MECCA/MOM
> as well as some evaluation against other chemical
> mechanisms, such as the more detailed benchmark MCM, over
> a range of atmospherically important chemical conditions.
> I realise that this is listed (and it is just a bullet
> pointed list) in the "Summary and outlook" section of the
> paper, but it is not clear if this work has been carried
> out, or is just a goal at the moment. In any case, some
> evaluation of the changes implemented in this version of
> CABBA/MECCA should be included in this paper before it is
> published in GMD.
```

To illustrate the changes in our model due to the updated MOM chemistry in MECCA, we have now performed 2 additional model runs, one using the simple MIM1 mechanism (Jöckel et al., 2016), and one using the MCM mechanism for isoprene and terpenes. The results have been added to Fig. 3 and to the text of the manuscript. Additional studies which analyze recent versions of MOM under a range of atmospherically important chemical conditions have been presented by Lelieveld et al. (2016), Cabrera-Perez et al. (2016), and Mallik et al. (2018).

```
> Perhaps mention "community model" in the title?
> (multi-purpose community atmospheric chemistry box
> model..)
```

We have added "community model" to the title.

```
> Introduction: A short descriptive definition of
> CABBA/MECCA and its history would be useful
```

We have added a description of CABBA/MECCA to the introduction.

```
> 2.1 Mainz Organic Mechanism (MOM) as the title
```

The title has been changed as requested.

```
> Page 2, Line 14  "explicit with a low degree of lumping"
> There for the scheme is not explicit!!
```

We never claimed that the *entire* scheme is explicit. We say that *most* of the oxidation scheme is explicit. Unfortunately, the referee cites only part of our sentence here.

> More explanation needed here with respect to "lumping".

We added the following text:

Lumping is used for some isomers with similar properties, e.g., the MOM species "LXYL" presents the sum of ortho-, meta- and para-xylene. All lumped species are marked by the prefix "L" in their names.

> Page 2, Line 16  The latest version of the MCM is v3.3.1,
> which is focused around a detailed update/evaluation to
> the MCM isoprene chemistry (Jenkin et al., Atmos. Chem.
> Phys., 15, 1143311459, 2015). MIM2 is based on MCMv3.1.
> Therefore, it would be useful to see how the updated MECCA
> model chemistry compares to MCMv3.3.1 in terms of isoprene
> chemistry under a range of representative conditions.

As already mentioned above, we have performed an additional model run, using the chemical mechanism of the current MCM for isoprene and terpenes. The results have been added to Fig. 3.

> Page 2, Line 25 - Given the last comment, what is the
> justification of not using the experimentally derived
> Crouse et al., (2012) 1,4H-shift kinetic data?

This issue has also been raised by referee 1. Please see our reply above.

> Page 2, Line 31  Updated aromatic chemistry. Has any
> evaluation of this updated chemistry been carried out
> against chamber data?

Not yet, but this is planned. One of us (DT) works at the Forschungszentrum Jülich where the SAPHIR chamber is located. An investigation of the $\alpha$- and $\beta$-pinene photooxidation by OH in the chamber has been presented very recently at the Atmospheric Chemical Mechanisms Conference in December 2018. MOM chemistry will be compared to these and other chamber results.

> How does it compare to other model aromatic atmospheric
> chemical mechanisms (the prediction of NO to NO2
> conversion and hence photochemical ozone formation, and
> chemical reactivity in various aromatics systems can vary
> significantly between different model chemistries.

This will be certainly an aspect of the mechanism evaluation that is intended in the future (see the reply above).

> Figure 1. Please list species by category i.e. alkanes,
> alkenes, aromatics, oxygenates etc

The figure has been rearranged as requested.

> Page 4. 2.1.1. VOC reactions with OH  This section is
> severally lacking in detail with respect to the updates
> actually carried out and the impact of such updates.
> Specifically: When were the IUPAC rate constants last
> updated? Referencing the IUPAC website datasheets as
> Atkinson et al., (2006) is somewhat out of date.

Several rate constants have already been updated according to the IUPAC website in those cases where the Atkinson et al. (2006) has been superseeded. Detailed information for every reaction is already available in the file mecca-nism.pdf in the supplement. Here, the rate constants and references can be found for all MECCA reactions. Specifically, the citation "Wallington et al. (2018)" refers to the IUPAC website. It is shown whenever we are using an up-dated rate constant from the IUPAC website which is different from Atkinson et al. (2006).

> The updated substituent factors form the original SAR of
> Kwok and Atkinson need to be shown here (in a table) so
> that they can be compared/contrasted to other SAR
> approaches and so the methodology is transparent to the
> community.

We have added the new Table 1 to the manuscript which shows the updated values.

> "No rigorous evaluation of the SAR has been conducted and
> estimation uncertainty is expected to be in the same range
> as the MCM" I strongly recommend that an evaluation of the
> updated SAR is carried out and compared to the original
> and other SAR approaches used in other models/chemical
> mechanisms.

We acknowledge the crucial importance of such evaluation. However, we think is out of the scope of the present manuscript. We plan to assess and possibly adopt the recent SAR by Jenkin et al. (2018) in the evaluation of MOM against SAPHIR experiments for a range of VOC.

```
> Page 4. "2.1.2 RO2 reaction with NOx"  Should be "with
> NOxy" as you also look at RO2 + NO3.
```

The name of the section has been changed to "RO$_2$ reactions with NO$_x$ and NO$_3$".

```
> Page 7. 2.1.4 RO2 permutation reactions, Line 11  "The
> rate expressions are not from the MCM, except" The rate
> expressions "not from the MCM" need to be defined and put
> into a Table here.
```

We have added a new table (Table 2) with details concerning the rate constants for this reaction category. We also show the MCM values and relevant references.

```
> Page 7. 2.1.5 Photo-induced reactions  Line 15 (and
> through-out) define photolysis rates as "j-values", i.e.
> j(NO2), and not "J-values" throughout.
```

We have changed to the lower-case notation as requested.

```
> Define "HPALD and "PACALD"
```

We define HPALD and PACALD as the conjugated unsaturated hydroperoxyenals from isoprene that have been originally proposed as the main source of OH recycling in isoprene chemistry. We have modified the text in section 2.1.5 accordingly.

```
> Figure 3 appears before it is mentioned in the text
```

The placement of the figures is indeed not correct yet. This will be corrected in collaboration with the Copernicus production office when the final two-column layout is produced.

```
> Page 8. 2.2 Other chemical mechanisms  some discussions,
> even brief ones about differences in the chemical schemes,
> would be useful
```

The following text has been added: *"All mechanisms are suitable for stratospheric as well as tropospheric calculations. They all include the chemistry of chlorine and bromine, and they all include isoprene and terpenes. However, only MCM, MOM and JAM002 treat some terpenes individually, e.g., pinene. The JAM002 mechanism is larger than CB05BASCOE and MOZART but small compared to MECCA with MOM. The very detailed MCM is the largest of all. More information about the chemical mechanisms is provided in the following sections."*

```
> Again, some evidence of evaluation of the updated
> MECCA/MOM chemistry against older versions and other
> detailed chemical mechanisms (such as the MCM) is needed
> here.
```

As mentioned above, a comparison of MOM and MCM has been added to the text and to Fig. 3.

```
> Page 11, Table 1. Needs much more explanation here. What
> is the colour coding in the table?
```

The following text has been added: *"The color coding [...] shows in which mechanism a species occurs. For example, orange is used for species which are included in the full mechanism and in s1 but not in s2 and s3."*

```
> Page 10, line 11  "from a global atmospheric chemistry
> simulation based on", describe the scenarios used in this
> simulation
```

We assume the referee refers to page 12 here, not page 10.

We agree that a better description is needed how we obtained the 30 sample points from the global model simulation. The revised text is now:

"Sample points were extracted from a global atmospheric chemistry simulation with a setup similar to that presented by Lelieveld et al. (2016). The chemical compositions were taken from several boxes at two altitudes (at the surface and at about 1 km). As we want the skeletal mechanism to perform well not only at typical concentrations of the targets but also when they are very high or very low, we picked boxes where the targets reach their minimum, average, or maximum concentrations, respectively. This resulted in the generation of 30 sample points (5 targets times (min/ave/max) times 2 altitudes), covering a wide range of values."

```
> Page 15, Line 13  Figure 5 appears after the References?!
```

The placement of the figures will be done from scratch after the layout has been changed to the two-column format. We will then ensure that the figures occur at the correct positions.

```
> Page 16. 4 MECCA in the MESSy modelling system  MESSy
> needs defining
```

We have now defined the acronym MESSy when it first occurs in the text, i.e., in the introduction.

```
> Page 20. Summary and outlook. Not actually a "summary" but
> a list of high level desirable goals. This should be
> re-written.
```

We admit that this section so far has been more "outlook" than "summary". We have added more text summarizing the new features of CAABA/MECCA.

**References**

Atkinson, R.: A structure-activity relationship for the estimation of rate constants for the gas-phase reactions of OH radicals with organic compounds, Int. J. Chem. Kinetics, 19, 799–828, `doi:10.1002/kin.550190903`, 1987.

Atkinson, R., Baulch, D. L., Cox, R. A., Crowley, J. N., Hampson, R. F., Hynes, R. G., Jenkin, M. E., Rossi, M. J., Troe, J., and IUPAC Subcommittee: Evaluated kinetic and photochemical data for atmospheric chemistry: Volume II – gas phase reactions of organic species, Atmos. Chem. Phys., 6, 3625–4055, `doi:10.5194/ACP-6-3625-2006`, 2006.

Cabrera-Perez, D., Taraborrelli, D., Sander, R., and Pozzer, A.: Global atmospheric budget of simple monocyclic aromatic compounds, Atmos. Chem. Phys., 16, 6931–6947, `doi:10.5194/acp-16-6931-2016`, 2016.

Crounse, J. D., Knap, H. C., Ørnsø, K. B., Jørgensen, S., Paulot, F., Kjaergaard, H. G., and Wennberg, P. O.: Atmospheric fate of methacrolein. 1. peroxy radical isomerization following addition of OH and $O_2$, J. Phys. Chem. A, 116, 5756–5762, `doi:10.1021/jp211560u`, 2012.

Jenkin, M. E., Valorso, R., Aumont, B., Rickard, A. R., and Wallington, T. J.: Estimation of rate coefficients and branching ratios for gas-phase reactions of OH with aliphatic organic compounds for use in automated mechanism construction, Atmos. Chem. Phys., 18, 9297–9328, `doi:10.5194/acp-18-9297-2018`, 2018.

Jöckel, P., Tost, H., Pozzer, A., Kunze, M., Kirner, O., Brenninkmeijer, C. A. M., Brinkop, S., Cai, D. S., Dyroff, C., Eckstein, J., Frank, F., Garny, H., Gottschaldt, K.-D., Graf, P., Grewe, V., Kerkweg, A., Kern, B., Matthes, S., Mertens, M., Meul, S., Neumaier, M., Nützel, M., Oberländer-Hayn, S., Ruhnke, R., Runde, T., Sander, R., Scharffe, D., and Zahn, A.: Earth System Chemistry integrated Modelling (ESCiMo) with the Modular Earth Submodel System (MESSy, version 2.51), Geosci. Model Dev., 9, 1153–1200, `doi:10.5194/gmd-9-1153-2016`, 2016.

Kwok, E. S. C. and Atkinson, R.: Estimation of hydroxyl radical reaction rate constants for gas-phase organic compounds using a structure-reactivity relationship: an update, Atmos. Environ., 29, 1685–1695, `doi:10.1016/1352-2310(95)00069-B`, 1995.

Lelieveld, J., Gromov, S., Pozzer, A., and Taraborrelli, D.: Global tropospheric hydroxyl distribution, budget and reactivity, Atmos. Chem. Phys., 16, 12 477–12 493, `doi:10.5194/acp-16-12477-2016`, 2016.

Mallik, C., Tomsche, L., Bourtsoukidis, E., Crowley, J. N., Derstroff, B., Fischer, H., Hafermann, S., Hüser, I., Javed, U., Keßel, S., Lelieveld, J., Martinez, M., Meusel, H., Novelli, A., Phillips, G. J., Pozzer, A., Reiffs, A., Sander, R., Taraborrelli, D., Sauvage, C., Schuladen, J., Su, H., Williams, J., and Harder,

H.: Oxidation processes in the eastern Mediterranean atmosphere: evidence from the modelling of $HO_x$ measurements over Cyprus, Atmos. Chem. Phys., 18, 10 825–10 847, `doi:10.5194/acp-18-10825-2018`, 2018.

Taraborrelli, D., Lawrence, M. G., Crowley, J. N., Dillon, T. J., Gromov, S., Groß, C. B. M., Vereecken, L., and Lelieveld, J.: Hydroxyl radical buffered by isoprene oxidation over tropical forests, Nature Geosci., 5, 190–193, `doi: 10.1038/NGEO1405`, 2012.

Wallington, T. J., Ammann, M., Cox, R. A., Crowley, J. N., Herrmann, H., Jenkin, M. E., McNeill, V., Mellouki, A., Rossi, M. J., and Troe, J.: IU-PAC Task group on atmospheric chemical kinetic data evaluation: Evaluated kinetic data, URL `http://iupac.pole-ether.fr`, 2018.

---

## Author Response (AR2)

**Replies to reviewer 1**

> This is a much improved version of the paper, and is now
> largely publishable as it is. On balance, I have recommended
> "minor revisions", but I think it is important that more
> information and discussion is provided for the one set of
> results that are presented (and maybe some additional
> results) - and I have referred to this as a "major comment"
> in my report below. This information and discussion could
> mainly be provided in the supplement, but I think needs to
> be there.
>
> This is a revised version of a paper accompanying the
> community release of version 4.0 of the CAABA/MECCA
> atmospheric chemistry box model, with the model code
> included as an electronic supplement. The authors have made
> a number of changes in response to reviewer comments, and
> the paper is substantially improved compared with the
> version that was originally submitted. In particular, the
> information provided about the treatment of the OH-initiated
> VOC chemistry is much clearer and more informative. The
> paper is therefore much closer to being publishable in GMD.
>
> I still have one major comment, which relates to the
> presented mechanism comparison. As indicated before, it is
> important that some results are presented, and I am pleased
> that the mechanism performance comparison provided by the
> authors is now more detailed, and with additional mechanisms
> included. However, despite the improvement in presentation
> clarity, the reader is left wondering what the main causes
> of the performance differences are. Are they because of the
> applied terpene speciation and the number of different
> terpenes treated in the different mechanisms? Are they due
> to differences in the treatment of isoprene chemistry?

As shown in Fig. 3, the results for isoprene are very similar for all mechanisms. This is because for the main reactions with ozone, OH and $NO_3$, all mechanisms use very similar rate constants:

| ozone | CB05BASCOE | 1.04E-14*EXP(-1995./TEMP) |
|-------|------------|----------------------------|
|       | JAM        | 7.860E-15*EXP(-1913./TEMP) |
|       | MCM        | 1.03E-14*EXP(-1995/TEMP)   |
|       | MOZART     | 1.050E-14*EXP(-2000./TEMP) |
|       | MIM1       | 7.86E-15*EXP(-1913./temp)  |
|       | MOM        | 1.03E-14*EXP(-1995./temp)  |
| OH    | CB05BASCOE | 2.7E-11*EXP(390./TEMP)     |
|       | JAM        | 2.700E-11*EXP(390./TEMP)   |
|       | MCM        | 2.70E-11*EXP(390/TEMP)     |
|       | MOZART     | 2.540E-11*EXP(410./TEMP)   |
|       | MIM1       | 2.54E-11*EXP(410./temp)    |
|       | MOM        | 2.7E-11*EXP(390./temp)     |
| $NO_3$ | CB05BASCOE | 3.15E-12*EXP(-450./TEMP)  |
|       | JAM        | 3.030E-12*EXP(-446./TEMP)  |
|       | MCM        | 3.15E-12*EXP(-450/TEMP)    |
|       | MOZART     | 3.030E-12*EXP(-446./TEMP)  |
|       | MIM1       | 3.03E-12*EXP(-446./temp)   |
|       | MOM        | 3.0E-12*EXP(-450./temp)    |

In contrast, the different treatment of terpenes in the mechanisms does indeed have an effect on the results. This is now explicated in the following text, which has been added to the manuscript:

> All mechanisms show a very similar decay of the initial isoprene because they all use similar rate constants for the main reactions of isoprene with ozone, OH and $NO_3$. In contrast, the results for the terpenes differ. In CB05BASCOE and MOZART, the rate constants for the lumped terpenes are taken from $\alpha$-pinene. In the other mechanisms, $\beta$-pinene (and other terpenes) are considered individually. Since $\beta$-pinene reacts with ozone much slower than $\alpha$-pinene, the explicit treatment of $\beta$-pinene in the mechanism leads to a slower decay of the terpenes than in the lumped mechanisms.

```
> In relation to the former point, it is not clear how the
> terpene speciation was assigned to the different mechanisms
> (and referring to MIM1/MIM2, you cannot initialize with 500
> pmol/mol of nothing!).
```

The following information about the terpene speciation has been added to the caption of Fig. 3:

> 500 pmol/mol of terpenes (MOM: 100 pmol/mol of $\alpha$-pinene, $\beta$-pinene, camphene, carene, and sabinene each; CB05BASCOE and MOZART: lumped terpenes; MIM1: no terpenes; MCM and JAM002: 200 pmol/mol $\alpha$-pinene and 300 pmol/mol $\beta$-pinene)

The MIM1 mechanism does not include terpenes at all. Unfortunately, this was not mentioned in the "Other chemical mechanisms" section. We have added this information now.

```
> I think a little more explanation would be useful in the
> manuscript, along with some more detailed supporting
> information in the supplement so that the reader can
> actually understand what was done and what the results
> actually mean.
```

The results for all species for all 6 mechanisms (MOM, CB05BASCOE, MOZART, MIM1, MCM and JAM002) are already available in the supplement. Unfortunately, we did not mention this in the text. We now provide additional information about the model runs in a README file and added the following text:

> *Details about these model runs and the results for all species are available in the testsuite/cams directory in the supplement.*

```
> Perhaps do three comparisons (i) isoprene alone; (ii)
> isoprene + a-pinene + b-pinene for MOM, MCM and JAM002;
> (iii) the existing comparison for all mechanisms except
> MIM1/MIM2, with an explanation of how the extra terpenes
> were assigned in the MCM and JAM002 simulations. This
> sequence of comparisons would help the reader to see the
> origin of the performance differences. The current
> illustration alone is of limited value and is
> insufficient.
```

We feel that performing and analyzing additional mechanism intercomparison simulations is beyond the scope of the current manuscript, which is a GMD "Model description paper", not a "Model evaluation paper". The mechanisms which are now included in CAABA/MECCA are not new; they have all been used in previous studies, e.g.: CB05BASCOE (Huijnen et al., 2016), MOZART (Emmons et al., 2010), JAM002 (Schultz et al., 2018). The aim of the current manuscript is to offer these mechanisms to the CAABA/MECCA model users so that they can choose the best mechanism for their own needs. The aim is not to perform a detailed model intercomparison. We agree that such an analysis would be very interesting and think that it could be the subject of a separate paper.

```
> I also have some minor and typographical comments, as follows:
>
> Page 1, line 1: Insert space between 4.0 and of.
```

This has been corrected.

```
> Page 2, line 10: The authors seem to have missed the point
> concerning my previous comment about the use of the terms
> VOCs and NMHCs  perhaps I was not as clear as I should
> have been. As shown in Fig. 1, the emitted VOCs are not only
> hydrocarbons (HCs), but oxygen- and nitrogen-containing
> organic compounds too. Therefore, CH4 + NMHCs = HCs or
> CH4 + NMVOCs = VOCs, but CH4 + NMHCs  VOCs. My original
> query was really why not use the term VOCs (or NMVOCs)
> throughout?
```

Thanks for the additional explanation. Indeed, we misunderstood the previous reviewer comment. We now only use the terms "VOCs" and "NMVOCs".

```
> Page 2, line 18: A very minor point, but I think primarily
> should be primary when referring to the emitted species
> (or could be omitted without changing the meaning).
> "Primary" is an adjective qualifying the species as distinct
> from secondary species. Primarily is an adverb (e.g. as
> used on page 12, line 12) describing the primary purpose of
> something, i.e. meaning mainly or for the most part.
```

The word "primarily" has been deleted.

```
> Page 4, line 35: It is rather than Its?
```

Changed as requested.

```
> Page 7, Fig. 3 caption: Is it MIM1 or MIM2?
```

Thanks for spotting this error. The caption has been corrected to "terpenes ([...]not included in MIM1)".

**Replies to reviewer 2**

```
> The paper has been improved with some useful additions.
>
> With respect to comments about the usefulness of the peer
> review publication component, please make sure that the
> supplemented CAABA/MECCA model code as well as the User
> Manual are appropriately sign posted in the paper.
```

Information how to obtain the CAABA/MECCA model code as well as the User Manual is already included in the "Code and data availability" section, as required for model description papers in GMD.

```
> Table 1 is useful. Would be good if you added the group rate
> coefficients calculated for 298K for comparison with the
> Kwok and Atkinson work.
```

We are unsure what kind of comparison the reviewer would like to make. To calculate the group rate constants (e.g., $k_p$, $k_s$, $k_t$), we use the same formulas as Kwok and Atkinson. Thus, our values at 298 K are also identical to those from Kwok and Atkinson. We think that adding them to Table 1 will not add any important information.

```
> In the "other chemical mechanisms" section, the discussion
> of the comparison of the different mechanism for a biogenic
> environment should come after you introduce/discuss the
> various mechanisms compared.
```

As suggested, the mechanism intercomparison has been moved into a new section (now: 2.3) after the introduction of the various mechanisms.

```
> How do the comparisons compare to wha was carried out in
> Emmerson and Evans Atmos. Chem. Phys., 9, 18311845, 2009?
```

Thanks for pointing out this reference to us. It is interesting to see a comparison made 10 years ago, partially based on predecessors of the mechanisms in this work: Their TOMCAT mechanism uses MIM1, their MOZART-2 is a predecessor of our MOZART, and their CBM-IV is a predecessor of our tropospheric chemistry in CB05BASCOE.

Like Emmerson and Evans, we also see quite different results for PAN when comparing the mechanisms. Regarding isoprene chemistry, however, all mechanisms of today produce very similar results. We now mention this in the text.

**References**

[revised manuscript text omitted]
 in- stance, one can distinguish oxidation generations: in re- ⁶⁰ actions with given oxidants the products become "pro- moted" to the tagging class of the next oxidation gener- ation. Another application of "class shifting" is quanti- fying the efficiency of recycling chains. In essence, such is the "online" implementation of the approach simi- ⁶⁵ lar to that of Lehmann (2004), with the number of tag- ging classes defining the maximum of the recycling se- quences it is possible to follow.

The range of MECCA-TAG applications was extended with new tagging setups/configurations: ⁷⁰

– Radiocarbon configurations, which facilitate simulating the ¹⁴C content in a desired set of species, including the routines for calculating abundances using conventional units like pMC (percent Modern Carbon).

– Hydrogen isotope chemistry: Now MECCA-TAG al- ⁷⁵ lows tracing pathways of H transfer between the species in the mechanism. Furthermore, D/H isotope chemistry (including relevant kinetic isotope effects for HOₓ and $C_1 - C_2$ chemistry) are included. The configuration and calculations of the composition transfer were extended ⁸⁰ with the possibility to specify isotope branching ratios necessary for the consistent D/H kinetics simulations. Both H transfer and D/H chemistry are currently eval- uated in stratospheric setups of CAABA (Frank et al., 2018). ⁸⁵

– O₂ clumped isotope chemistry: simulation of non- stochastic distributions of ¹⁸O¹⁸O and ¹⁷O¹⁸O isotopo- logues ($\Delta_{36}$ and $\Delta_{35}$ signatures) resulting from O($^3$P)- mediated temperature-dependent isotope exchange ki- netics. ⁹⁰

[revised manuscript text omitted]